# TPL2 kinase activity regulates microglial inflammatory responses and promotes neurodegeneration in tauopathy mice

Yuanyuan Wang[1]*, Tiffany Wu[1], Ming-Chi Tsai[1], Mitchell G Rezzonico[2], Alyaa M Abdel-Haleem[3], Luke Xie[4], Vineela D Gandham[4], Hai Ngu[5], Kimberly Stark[1], Caspar Glock[2], Daqi Xu[6], Oded Foreman[5], Brad A Friedman[2], Morgan Sheng[1,7], Jesse E Hanson[1]*

[1]Department of Neuroscience, Genentech Inc, South San Francisco, United States; [2]Department of OMNI Bioinformatics, Genentech Inc, South San Francisco, United States; [3]Computational Science & Exploratory Analytics, Roche IT, Hoffmann-La Roche Limited, Mississauga, Canada; [4]Department of Translational Imaging, Genentech Inc, South San Francisco, United States; [5]Department of Pathology, Genentech Inc, South San Francisco, United States; [6]Department of Immunology, Genentech Inc, South San Francisco, United States; [7]Stanley Center for Psychiatric Research, Broad Institute of MIT and Harvard, Cambridge, United States

*For correspondence:
wangy111@gene.com (YW);
hanson.jesse@gene.com (JEH)

**Abstract** Tumor progression locus 2 (TPL2) (MAP3K8) is a central signaling node in the inflammatory response of peripheral immune cells. We find that TPL2 kinase activity modulates microglial cytokine release and is required for microglia-mediated neuron death in vitro. In acute in vivo neuroinflammation settings, TPL2 kinase activity regulates microglia activation states and brain cytokine levels. In a tauopathy model of chronic neurodegeneration, loss of TPL2 kinase activity reduces neuroinflammation and rescues synapse loss, brain volume loss, and behavioral deficits. Single-cell RNA sequencing analysis indicates that protection in the tauopathy model was associated with reductions in activated microglia subpopulations as well as infiltrating peripheral immune cells. Overall, using various models, we find that TPL2 kinase activity can promote multiple harmful consequences of microglial activation in the brain including cytokine release, iNOS (inducible nitric oxide synthase) induction, astrocyte activation, and immune cell infiltration. Consequently, inhibiting TPL2 kinase activity could represent a potential therapeutic strategy in neurodegenerative conditions.

## Editor's evaluation

In this study, the authors provide important findings supporting a key role for TLP2 as a regulator of neurotoxic and pro-inflammatory cytokine and chemokine release following acute and chronic neuroinflammation. They provide convincing data supporting that the abrogation of TPL2 kinase activity ameliorates disease pathogenesis in a mouse model of tauopathy. This manuscript will be of broad interest to readers in the fields of neuroimmunology and neurodegenerative disease who are interested in the pathogenic effects of innate immune signaling pathways in disease.

## Introduction

Activation of microglia, the tissue resident macrophages of the brain, occurs in numerous neurodegenerative conditions including Alzheimer's disease (AD) and frontotemporal dementia (FTD), where microglia are activated in response to amyloid and tau pathology. While microglia activation

may facilitate beneficial functions, such as clearance of amyloid plaques, in the early stages of AD, chronic microglia activation could contribute to disease progression via cytokine release and other harmful functions during later stages of disease (*Bright et al., 2019*; *Hansen et al., 2018*; *Leng and Edison, 2021*). A pathogenic role for inflammatory cytokines in neurodegeneration is supported by the increased risk for AD in people with rheumatoid arthritis (RA), psoriasis, and other inflammatory diseases, and a significantly reduced incidence of AD for those patients that received treatment with TNF blocking agents (*Zhou et al., 2020*). Several aspects of neuroinflammation that have been observed in neurodegeneration patient brains have been studied in preclinical models and found to have deleterious effects. This includes studies in AD and FTD models that identify harmful effects of activated astrocytes (*Li et al., 2019*), NLRP3 inflammasome activation (*Heneka et al., 2013*; *Ising et al., 2019*), complement pathway activation (*Dejanovic et al., 2018*; *Hong et al., 2016*; *Wu et al., 2019*), and infiltration of T-cells (*Gate et al., 2020*; *Laurent et al., 2017*; *Merlini et al., 2018*).

Given such diverse mediators of the detrimental effects of neuroinflammation and microglia activation, targeting a central regulator of inflammatory function represents an attractive therapeutic approach. We therefore focused on tumor progression locus 2 (TPL2), a kinase that is expressed in both adaptive and innate immune cells in the peripheral system and plays central roles in regulating inflammatory responses. TPL2 is a mitogen-activated protein kinase kinase kinase (MAP3K) downstream of TNFR, IL1R, and Toll-like receptors. It is maintained in an inactive complex with ABIN-2 (A20-binding inhibitor of NF-κB 2) and p105, and is released upon activation. TPL2 activation is a critical node of the inflammatory response with activation leading to changes in inflammatory gene transcripts via ERK1/2 and p38 activation, resulting in amplification of inflammatory responses in peripheral immune cells (*Xu et al., 2018*). Preclinical studies using genetic manipulation of TPL2 in mice have found that TPL2 deficiency provides beneficial effects in multiple inflammatory disease models, such as inflammatory bowel disease, psoriasis, pancreatitis, and multiple sclerosis (*Senger et al., 2017*; *Sriskantharajah et al., 2014*; *Xiao et al., 2014*; *Xu et al., 2018*), and TPL2 is being pursued as a target for treating inflammatory conditions including RA and psoriasis (*Zarrin et al., 2021*).

To evaluate a potential role for TPL2 kinase activity in microglia during neuroinflammation, we used TPL2 inhibitors and TPL2 kinase dead (TPL2-KD) mice (D270A mutation). TPL2-KD mice were chosen instead of TPL2 knockout (KO) mice because TPL2 is required to maintain stable expression of its interacting partner ABIN-2 and TPL2 KO mice have dramatically reduced ABIN-2 levels, which could confound interpretations regarding the role of TPL2 kinase activity (*Sriskantharajah et al., 2014*; *Webb et al., 2019*). We first examined the function of TPL2 in microglial inflammation using in vitro models, then in acute neuroinflammation settings, and finally in a tauopathy model of chronic neurodegeneration. These experiments indicate that TPL2 regulates microglia inflammatory activity and damage to neurons, and that blocking TPL2 function can be neuroprotective. Measurement of the cellular changes that occurred in the brains of the tauopathy mouse model with TPL2-KD along with reduced neurodegeneration and normalization of behavior reveal multiple potential beneficial aspects of preventing TPL2 kinase activity.

## Results

### TPL2 modulates inflammatory responses of microglia

TPL2 is expressed in both innate and adaptive immune cells in the peripheral immune system to regulate inflammatory responses (*Xu et al., 2018*). To examine the expression of TPL2 in the brain under resting and disease conditions, we analyzed published datasets as well as datasets we recently generated. RNA sequencing (RNA-seq) analysis of FACS-purified cells from human (*Srinivasan et al., 2020*) and mouse (*Zhang et al., 2014*) cortex revealed that TPL2 expression is enriched in brain myeloid cells and endothelial cells (*Figure 1—figure supplement 1A*). In an acute neuroinflammation study, in which mice received intraperitoneal injection of the endotoxin lipopolysaccharide (LPS), TPL2 expression was highly induced in both bulk brain tissue (GSE196401) and sorted microglia (*Srinivasan et al., 2016*). We also found that TPL2 gene expression was upregulated in bulk tissue (GSE186414) and sorted microglia from the P301S tauopathy mouse model (*Friedman et al., 2018*; *Figure 1—figure supplement 1B*). Western blot analysis of TPL2 protein levels in the cortex of Alzheimer's disease (AD) patients showed an increase compared to healthy control brains (*Figure 1—figure supplement 1C*).

We first used mouse primary microglia cultures to investigate the TPL2 signaling cascade in this brain cell type. Upon LPS stimulation, phosphorylation of ERK and p38 was dramatically increased, and this increased phosphorylation was significantly dampened in the presence of the TPL2 small molecule inhibitors G-767 and G-432 (*Figure 1A and B*; *Figure 1—figure supplement 2A*). In contrast, as expected, LPS-induced nuclear factor-kappa B (NF-κB) pathway activation, as measured by elevation of phospho-p65, was not affected by TPL2 inhibition. Of note, the TPL2 inhibitors did not affect baseline phosphorylation of ERK, p38 or p65 without stimulation (*Figure 1A, B*). These results are consistent with TPL2 mediating mitogen-activated protein kinase (MAPK) signaling, but not NF-κB signaling, in microglia. It is well established that TPL2 plays an important role in regulating cytokine production by myeloid cells in the peripheral immune system via the MAPK signaling pathway (*Arthur and Ley, 2013*). Consistently, using multiplex immunoassays to measure cytokine release by microglia, we found that LPS-induced production of inflammatory cytokines, such as TNFα, IL-1α, IL-6, and CXCL1, was greatly reduced in the presence of TPL2 inhibitors (*Figure 1C*, *Figure 1—figure supplement 2B*, *Figure 1—figure supplement 3A*). Moreover, when we analyzed cytokine release by microglia cultured from TPL2-KD mice, there was dramatically decreased LPS-induced cytokine production (*Figure 1D* and *Figure 1—figure supplement 3B*). Overall, this data indicates that TPL2 is a key regulator of the microglial MAPK signaling pathway and proinflammatory cytokine responses, and thus could play an important role in modulating neuroinflammation in the brain.

## Inhibition of TPL2 kinase activity protects against neuronal death in a co-culture model

Activated microglia can be neurotoxic by releasing factors such as proinflammatory cytokines and reactive oxygen species (*Glass et al., 2010*; *Wang et al., 2015*). Since we found that TPL2 can modulate proinflammatory responses in microglia, we next examined if targeting TPL2 could dampen neurotoxic functions of activated microglia. To investigate this, we used an in vitro microglia and neuron co-culture system and tested pharmacological and genetic inhibition of TPL2 kinase activity. Stimulation of microglia and neuron co-cultures with LPS + interferon gamma (IFNγ), a stimulus reported to induce strong microglia activation (*Straccia et al., 2011*), resulted in almost complete neuronal loss as measured by MAP2 immunostaining (*Figure 2A and B*). Strikingly, when the TPL2 inhibitor G-767 was added to the co-culture, the neuronal loss induced by the LPS + IFNγ stimulation was nearly fully rescued. In addition, when WT neurons were co-cultured with microglia isolated from TPL2-KD mice, stimulation-induced neuronal loss was also rescued (*Figure 2A and B*). Of note, when neurons were cultured alone and stimulated with LPS + IFNγ or co-cultured with microglia at 1:1 ratio without stimulation, no obvious neuronal loss was observed (*Figure 2—figure supplement 1*).

We next wanted to understand the mechanism of neuronal death and the neuroprotective effect of TPL2 kinase inhibition in this co-culture system. Several groups have shown that LPS + IFNγ stimulation can induce dramatic increases of inducible nitric oxide synthase (iNOS) expression in microglia and subsequent release of nitric oxide (NO), which is neurotoxic (*Papageorgiou et al., 2016*; *Sheng et al., 2011*; *Straccia et al., 2011*). Consistent with this mechanism, we found that when primary microglia cultures were stimulated with LPS + IFNγ, iNOS protein levels were increased and accumulated over time (*Figure 2C and D*). TPL2 small molecule inhibition significantly reduced iNOS protein in stimulated microglia (50% reduction) and stimulated microglia isolated from TPL2-KD mice also showed a dramatic reduction of iNOS (85% reduction) (*Figure 2C–E*). To determine if NO plays a role in neuronal death in the co-cultures, we utilized the iNOS inhibitor 1400W. Treatment with the iNOS inhibitor almost fully rescued the neuronal loss (*Figure 2F, G*), indicating that NO production by activated microglia is a key contributor to neuronal death in this co-culture system.

## TPL2 reduces acute neuroinflammation in an in vivo LPS injection model

We next examined the role of TPL2 in vivo using an acute neuroinflammation model. We injected WT or TPL2-KD mice with either phosphate-buffered saline (PBS) or LPS (10 mg/kg i.p.) and collected brain tissue 24 hr later for bulk RNA-seq analysis. Using previously defined CNS myeloid gene modules (*Friedman et al., 2018*), we found that LPS induced significantly increased expression of disease-associated microglia (DAM) genes (e.g. *Il1r1*, *Axl*) in WT mice, whereas this microglia activation response was attenuated in LPS-treated TPL2-KD mice (*Figure 3A, D, and E*). Homeostatic microglia gene expression (e.g. *Tmem119*, *Cx3cr1*) was downregulated by LPS in WT mice, but

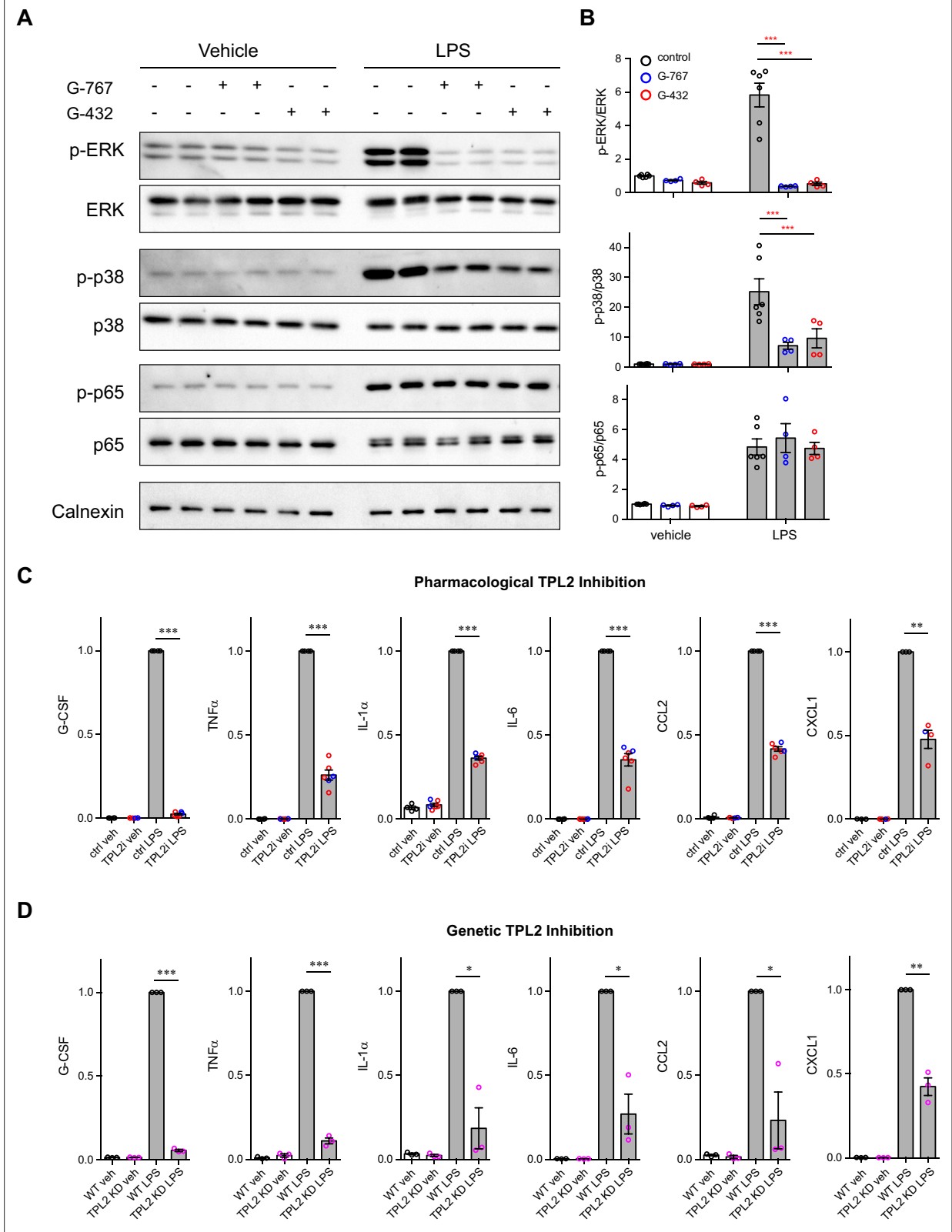

**Figure 1.** Tumor progression locus 2 (TPL2) modulates mitogen-activated protein kinase (MAPK) signaling and cytokine production by microglia. (**A**) Representative western blots of phospho-ERK, total ERK, phospho-p38, total p38, phospho-p65, total p65, and calnexin (loading control) in lysates from primary microglia cultures treated with vehicle or lipopolysaccharide (LPS) with or without TPL2 inhibitors (2 μM). (**B**) Quantification of western blot data as shown in (**A**). Phosphorylated proteins were first normalized to the corresponding total proteins, and then the normalized values were plotted relative

*Figure 1 continued on next page*

*Figure 1 continued*

to the average values from control samples (no LPS, no TPL2 inhibitor) n=4–6. Data are represented by mean ± SEM. ***, p<0.001, two-way ANOVA with Tukey's multiple comparisons test. (C and D) Quantification of cytokine measurements from the supernatants of WT primary microglia with or without TPL2 inhibitor (C) or microglia isolated from WT mice or TPL2 kinase dead (TPL2-KD) mice (D) under control or LPS induction (24 hr incubation) conditions. Cytokines highlighted here were selected based on significant LPS induction that was reduced by TPL2 inhibition (pharmacological or genetic) by >50%. Additional cytokine measurements are provided in *Figure 1—figure supplement 3*. Cytokine measurements were normalized to average values of WT microglia treated with LPS. Data from TPL2 inhibitor G-432 (red circle) and G-767 (blue circle) were combined in (C), n=4–6. n=3 for (d). Data are represented by mean ± SEM. *, p<0.05, **, p<0.01, ***, p<0.001, one-way ANOVA with Tukey's multiple comparisons test.

The online version of this article includes the following source data and figure supplement(s) for figure 1:

**Source data 1.** Full images of *Figure 1A* western blots are available.

**Figure supplement 1.** Tumor progression locus 2 (TPL2) is upregulated in tauopathy mice and Alzheimer's disease (AD) patients.

**Figure supplement 1—source data 1.** Full images of *Figure 1—figure supplement 1C* western blots are available.

**Figure supplement 2.** Tumor progression locus 2 (TPL2) inhibitor structures and potency dose-response curves.

**Figure supplement 3.** Cytokine and chemokine release by microglia after lipopolysaccharide (LPS) stimulation.

relatively unchanged in LPS-treated TPL2-KD mice (*Figure 3C, D, and E*), consistent with TPL2-KD microglia being in a less activated state after stimulation. Interestingly, an increase in microglia proliferation genes by LPS in WT mice was also normalized in TPL2-KD mice (*Figure 3E*).

In addition to directly affecting neurons, microglia have also been shown to indirectly cause neurotoxicity by activating astrocytes. Cytokines released by microglia in response to LPS stimulation can induce the transition of astrocytes from a resting state to an activated/reactive state that can be neurotoxic (*Liddelow et al., 2017*). Consistent with a reduction of microglia activation of astrocytes, LPS-induced activated astrocyte gene expression (e.g. *Cd44, Srgn*) was also attenuated in TPL2-KD mice (*Figure 3B, D, and E*). In parallel to the RNA-seq measurements, we also measured brain cytokine levels and found that several cytokines such as IL-1α, IL-6, and CXCL1 that were elevated in LPS-treated WT mouse brains had significantly lower levels in TPL2-KD mouse brains (*Figure 3F*). Since astrocytes also express TPL2, especially under stimulated conditions (*Figure 1—figure supplement 1*), we investigated if TPL2 could directly regulate astrocyte activation state using primary astrocyte cultures. In these experiments astrocytes were stimulated by either LPS (*Figure 3—figure supplement 1A*) or a cytokine cocktail (TNFα+IL-1α) (*Figure 3—figure supplement 1B*), with or without TPL2 inhibition. Upon stimulation, primary astrocytes released increased amounts of various cytokines. In contrast to the broader TPL2 dependence of cytokine release by cultured microglia (*Figure 1C*, *Figure 1—figure supplement 3A*), only a much more restricted set of cytokines exhibited TPL2 dependence in cultured astrocytes. Moreover, TPL2-dependent activated astrocyte genes identified in the in vivo LPS study (*Figure 3B*) exhibited much less TPL2-dependent activation in cultured astrocytes (*Figure 3—figure supplement 1C and D*). These results suggest that the TPL2-dependent astrocyte activation we observed in vivo following LPS injection was probably largely contributed to indirectly via the function of TPL2 in microglia, but there was also potentially some contribution of cell-autonomous function of TPL2 in astrocytes. Collectively, these results indicate that TPL2 plays a key role in microglia and astrocyte activation in vivo, and show that acute neuroinflammation can be reduced by ablation of TPL2 kinase activation.

We next investigated if TPL2 kinase deficiency could protect against neuronal damage in acute injury models. We first tested the mouse optic nerve crush (ONC) model. However, although we observed moderate reduction of microgliosis in TPL2-KD mice after ONC, we did not see rescue of retinal ganglion cell loss (*Figure 3—figure supplement 2A*). Additionally, we also investigated the role of TPL2 in another acute model, the rat transient middle cerebral artery occlusion (tMCAO) model of stroke. Similarly, despite significantly reduced microgliosis, we did not see beneficial effects in TPL2-KD rats in terms of body weight loss, behavioral deficits, and lesion volume measured by MRI after tMCAO (*Figure 3—figure supplement 2B–H*). These results highlight that reduced neuroinflammation with TPL2-KD may not be sufficient for neuroprotection in cases of severe acute tissue damage.

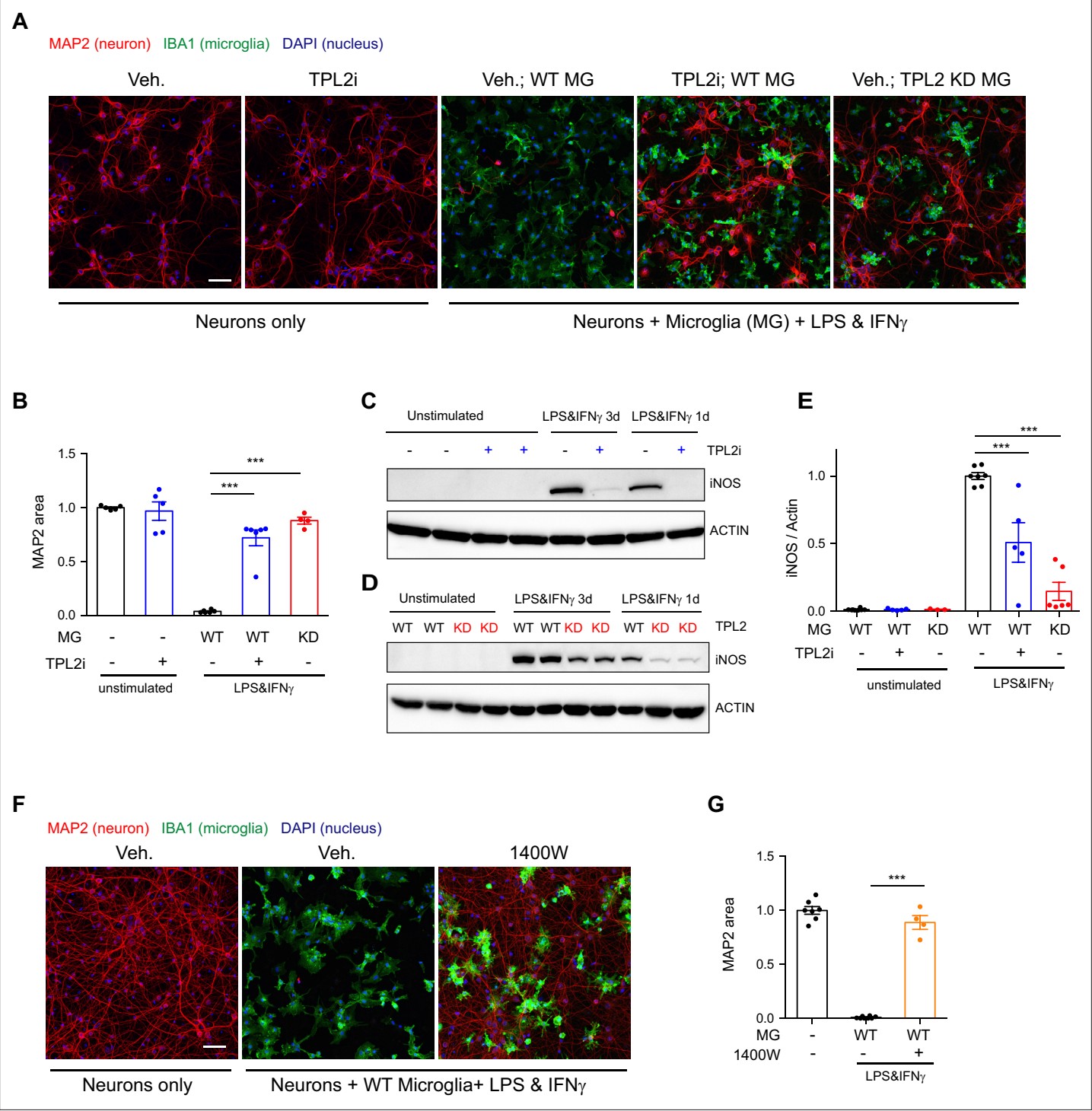

**Figure 2.** Tumor progression locus 2 (TPL2) inhibition rescues neurodegeneration in a co-culture model. (**A**) Representative images of immunostaining for MAP2 (red), IBA1 (green), and DAPI (blue) in mouse cortical neuronal cultures or mouse cortical neurons co-cultured with mouse microglia as indicated. Cell number ratio in co-cultures is 1:1. Neurons were always cultured from WT mice and microglia were either from WT mice or TPL2 kinase dead (TPL2-KD) mice as indicated. Scale bar, 50 μm. (**B**) Quantification of MAP2 staining positive area within the image field as a readout of surviving neurons in the culture. MAP2 area is shown as normalized relative to the average value of the control condition (no microglia, no stimulation). n=5–6 per condition. (**C** and **D**) Representative immunoblots of iNOS (inducible nitric oxide synthase) from primary microglia cultures with treatments as indicated. Comparison was done either between WT microglia treated with and without TPL2 inhibitor G-767 (**C**), or between microglia isolated from WT mice and TPL2-KD mice (**D**). (**E**) Quantification of western blot data in (**C**) and (**D**). iNOS signal was normalized to actin signal, and the normalized values were plotted relative to the average value of the lysates from WT microglia treated with lipopolysaccharide (LPS) and interferon gamma. n=3–8.

*Figure 2 continued on next page*

Figure 2 continued

(**F**) Representative images of immunostaining of MAP2 (red), IBA1 (green), and DAPI (blue) in mouse cortical neuronal cultures or mouse cortical neurons co-cultured with mouse microglia as indicated. Cell number ratio in co-cultures is 1:1. Scale bar, 50 µm. (**G**) Quantification of MAP2 staining is shown as in (**B**). n=4–8. For all bar graphs, data are represented by mean ± SEM. \*\*\*, p<0.001, one-way ANOVA with Tukey's multiple comparisons test.

The online version of this article includes the following source data and figure supplement(s) for figure 2:

**Source data 1.** Full images of *Figure 2C, D* western blots are available.

**Figure supplement 1.** Stimulation of neurons cultured without microglia or co-culture of neurons and microglia without stimulation does not result in neuronal loss.

## Neuroinflammation in TauP301S tauopathy model is attenuated in TPL2-KD mice

To determine if TPL2 contributes to neuroinflammation and neurodegeneration in a chronic disease setting, we crossed TPL2-KD mice with TauP301S tauopathy mice. These transgenic mice overexpress human mutant Tau and develop pathology marked by phosphoTau accumulation, neuroinflammation, and brain atrophy as the mice age, especially in the hippocampus (*Wu et al., 2019*; *Yoshiyama et al., 2007*). We examined astrogliosis and microgliosis by analyzing the microglia marker Iba1 and the astrocyte marker GFAP in 9-month-old male mice and 11-month-old female mice, as female mice develop the Tau pathology slower compared with the male mice in this mouse line based on our previous experience (*Wu et al., 2019*). TPL2-KD did not affect the % area of the brain covered by Iba1 or GFAP in non-transgenic mice. In TauP301S mice, the Iba1 and GFAP signals were greatly increased reflecting gliosis, and TPL2 kinase deficiency slightly ameliorated this gliosis (*Figure 4A, B*). The effect was observed with either whole brain section analysis or hippocampal region analysis. Interestingly, phospho-Tau pathology (measured by AT8 staining) was also somewhat reduced in TauP301S;T-PL2-KD mice (*Figure 4A, B*), potentially reflecting the interplay between neuroinflammation and Tau pathology (*Didonna, 2020*; *Maphis et al., 2015*). Consistent with the gliosis changes, the levels of several cytokines and chemokines that were significantly increased in TauP301S mouse brains were blunted in TauP301S;TPL2-KD brains, including CXCL9, CXCL10, IL6, and IFNγ (*Figure 4C*).

## Single-cell RNA-seq analysis reveals reduced peripheral immune cell infiltration in P301S;TPL2-KD mice

To gain further insight into the effect of TPL2-KD in the P301S brain, we performed single-cell RNA sequencing (scRNA-seq) of the hippocampus, a brain region where pathology is most prominent in the TauP301S model. Dimensional reduction by Seurat clustering and interpretation of clusters using previously established gene sets (*Friedman et al., 2018*; *Lee et al., 2021b*; *Zeisel et al., 2018*) resulted in annotation of 13 cell types (*Figure 5A*, *Figure 5—figure supplement 1*, *Supplementary file 1*). To investigate how Tau pathology and TPL2 kinase activity affected the relative abundance of various cell types, we examined cellularity across genotypes (*Figure 5—figure supplement 2*). P301S mice showed increases in relative abundance of several cell types, including microglia, T-cells, and a cluster we initially labeled as 'mixed immune cells' (*Figure 5A*, *Figure 5—figure supplement 1B*, *Figure 5—figure supplement 2*). Further analysis of the mixed immune cells cluster showed that it contained a unique cell type with expression of distinct genes compared to microglia or T-cells (e.g. *Plbd1, Mgl2, Clec9a*) (*Figure 5—figure supplement 3A*). Subclustering of the mixed immune cells identified this unique cell type as a distinct cluster from microglia or T-cells (*Figure 5—figure supplement 3*), and mapping onto ImmGen cell type data (*Heng and Painter, 2008*) revealed this population corresponded to dendritic cells (*Figure 5—figure supplement 3*). Like microglia and T-cells, dendritic cells also showed an elevation in P301S brains (*Figure 5C*; *Figure 5—figure supplement 4*, C21). Strikingly, while TPL2-KD did not have a significant effect on the proportion of any of the 13 cell types in non-transgenic mice, and did not alter the proportion of most cell types in P301S brains, TPL2-KD significantly reduced the abundance of T-cells and dendritic cells in P301S brains (*Figure 5B and C*; *Figure 5—figure supplement 4*). This indicates that P301S;TPL2-KD mice have reduced infiltration of peripheral immune cells compared to P301S mice. To corroborate the scRNA-seq results, we also performed T-cell immunohistochemistry (IHC) using fixed brain tissue sections. T-cell number in the fimbria area (the area that has the most robust T-cell abundance in previous studies of TauPS2APP mice; *Lee et al., 2021b*) was increased in TauP301S mouse brains, and the increase was normalized

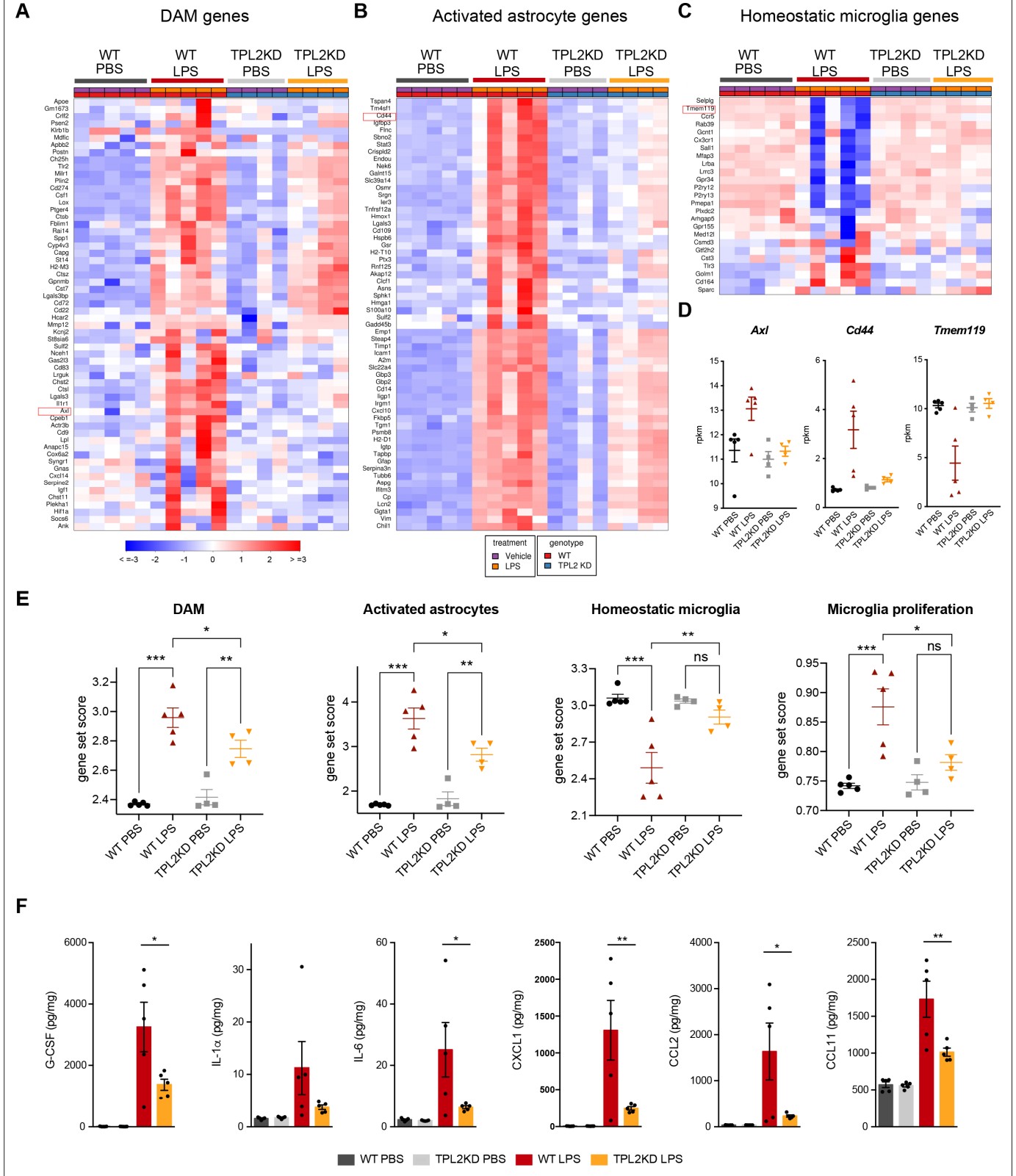

**Figure 3.** Tumor progression locus 2 kinase dead (TPL2-KD) mice have attenuated gene expression changes and cytokine/chemokine response after lipopolysaccharide (LPS) injection. (**A**) Heatmap showing z-scores of the top 60 DAM genes (disease-associated microglia genes) in the brains of WT or TPL2-KD mice harvested 24 hr post intraperitoneal injection with phosphate-buffered saline (PBS) or LPS (10 mg/kg). The top 60 DAM genes were taken from the list in *Friedman et al., 2018*, and were selected based on the ranking of z-scores of these genes in WT mice injected with LPS. (**B**) Heatmap

*Figure 3 continued on next page*

*Figure 3 continued*

showing z-scores of the top 60 activated astrocyte genes taken from the list in *Wu et al., 2019* (both A1 and A2 astrocyte genes were included). The top 60 genes were also selected based on the ranking of z-scores of these genes in WT mice injected with LPS. (**C**) Heatmap showing z-scores of homeostatic microglia genes from *Friedman et al., 2018*. (**D**) Examples of DAM genes (*Axl*), astrocyte-activated genes (*Cd44*), and homeostatic microglia genes (*Tmem119*). (**E**) Geneset score expression of DAM genes, activated astrocyte genes, homeostatic microglia genes, and microglia proliferation genes in mouse brain samples with different treatments as indicated. n=4–5 animals per condition. (**F**) Cytokine and chemokine levels in mouse brain lysates from the same mice as in (**A–E**) measured using a bead-based multiplex assay. Cytokine levels were normalized to total protein concentration in brain lysates and are shown as pg/mg. Only analytes that were above the detection level, changed after LPS injection, and exhibited a difference between WT and TPL2-KD mice are shown. Data are represented by mean ± SEM. *, $p<0.05$, **, $p<0.01$, ***, $p<0.001$, one-way ANOVA with Tukey's multiple comparisons test.

The online version of this article includes the following figure supplement(s) for figure 3:

**Figure supplement 1.** Cell-autonomous role of tumor progression locus 2 (TPL2) in activated astrocytes.

**Figure supplement 2.** Effect of tumor progression locus 2 kinase dead (TPL2-KD) in acute injury models: optic nerve crush (ONC) and stroke.

by TPL2-KD (*Figure 5—figure supplement 2B and C*). Given the presence of T-cells in the brains of human AD patients and preclinical disease models, and the beneficial effects of T-cell depletion in mouse disease or injury models (*Chen et al., 2023*; *Gate et al., 2020*; *Laurent et al., 2017*; *Merlini et al., 2018*; *Togo et al., 2002*), the decreased presence of peripheral immune cells in TauP301SxT-PL2-KD mouse brains could have beneficial effects.

Subclustering of the T-cells identified two clusters of CD4+ (CD4.1–2) cells and four clusters of CD8+ cells (CD8.1–4), with CD8+ cells being the dominant T-cell population (*Figure 5—figure supplement 5A, B*). Cellularity analysis showed that both CD4+ and CD8+ cell numbers were increased in P301S brains, and TPL2-KD significantly reduced the abundance of both cell types (*Figure 5—figure supplement 5C*). When the relative abundance was compared within the T-cell population, CD8.4, the largest T-cell subcluster, was decreased in P301S;TPL2-KD mouse brains (*Figure 5—figure supplement 5D*). CD8.4 cells expressed some marker genes of exhausted T-cells, such as *Nr4a2* and *Pdcd1*, which are usually induced by chronic/persistent exposure to antigens and/or inflammation (*Odagiu et al., 2020*; *Wherry, 2011*; *Wherry and Kurachi, 2015*; *Yi et al., 2010*).

## Normalization of microglial gene expression changes in P301S;TPL2-KD mice

To investigate the genotype-dependent gene expression changes in various cell types, we performed pseudo-bulk analysis for each cell type in this study (*Figure 6—figure supplement 1*). Comparing TauP301S with non-transgenic mice, there were many differentially expressed genes (DEGs) across multiple cell types, such as astrocytes, microglia, and oligodendrocytes (*Figure 6—figure supplement 1A*). However, within the TauP301S group, when comparing TPL2-KD to TPL2-WT mice, DEGs were only detected in microglia (*Figure 6A*), consistent with TPL2 being most highly expressed by microglia in the mouse brain (*Figure 1—figure supplement 1*). Analysis of the protein-coding DEGs showed that the most upregulated genes in P301S microglia were consistently upregulated to a lesser extent in P301S;TPL2-KD microglia, including cytokines like *Tnf* and multiple immediate early genes (IEGs) such as *Egr1* (*Figure 6B, C*).

To further analyze microglia phenotypes, the ~55,000 microglia in the study were subclustered into 22 clusters (*Figure 6D*). As expected, clusters of microglia annotated as neurodegeneration-related and interferon-related were elevated, and clusters annotated as resting/homeostatic were decreased in TauP301S mice (*Figure 6—figure supplements 1 and 2*). However, the abundance of these clusters in P301S mice were not altered by TPL2-KD (*Figure 6—figure supplements 1 and 2*). On the other hand, a subcluster of microglia (C11) expressing high levels of IEGs (e.g. *Fos, Egr1-3, Atf3*) (*Figure 6—figure supplement 1E*) was elevated in P301S mice, and this elevation was ameliorated by TPL2-KD (*Figure 6E*). Calculation of IEG gene set score for every defined cell type (pseudo-bulk) showed IEG expression was up in astrocytes, microglia, oligodendrocytes, OPCs, and VSMCs, but the upregulation was only normalized by TPL2-KD in microglia (and to some extent in VSMCs, although not statistically significant, *Figure 6—figure supplement 1G*). As IEG expression can be regulated by MAPK signaling pathways and TPL2 is a key regulator of MAPK signaling (*Bahrami and Drabløs, 2016*; *O'Donnell et al., 2012*), these results are consistent with a specific role for TPL2 in TauP301S microglia. Another subcluster of microglia (C20) that was elevated in P301S and normalized

**Figure 4.** Tumor progression locus 2 kinase dead (TPL2-KD) partially rescues neuroinflammation in TauP301S mouse brain. (**A**) Representative images showing immunostaining for Iba1, GFAP, and AT8 in mouse hemi-brains from 9-month-old male mice with genotypes as indicated. Scale bar, 500 μm. (**B**) Quantification of percentage of Iba1, GFAP, AT8 positive area within mouse hemi-brains and specifically within the hippocampus as indicated. Analysis shown are combined data from 9-month-old male mice and 11-month-old female mice. Each dot represents one animal. (**C**) A subset of cytokines or chemokine levels in mouse hippocampal lysates measured using bead-based mouse cytokine multiplex assay. Cytokine levels were normalized to total protein concentration in brain lysates and are shown as pg/mg. Data shown include protein levels from both 9-month-old male mice and 11-month-old female mice. For all bar graphs, data are represented by mean ± SEM. *, p<0.05, ***, p<0.001, one-way ANOVA with Tukey's multiple comparisons test.

by TPL2-KD in the P301S mice expressed high levels of MHC (major histocompatibility complex) class II genes (e.g. *Cd74, H2-Aa, H2-Ab1*, etc.), which are usually expressed by antigen presenting cells and are important for the generation of immune responses via interaction with T-cells (*Neefjes et al., 2011*; *Roche and Furuta, 2015*; *Figure 6F*, *Figure 6—figure supplement 1F*). The reduction of P301S MHCII expressing microglia by TPL2-KD in P301S mice combined with the reduction of infiltrating T-cells and dendritic cells points to a normalization of adaptive immune function in P301S brains.

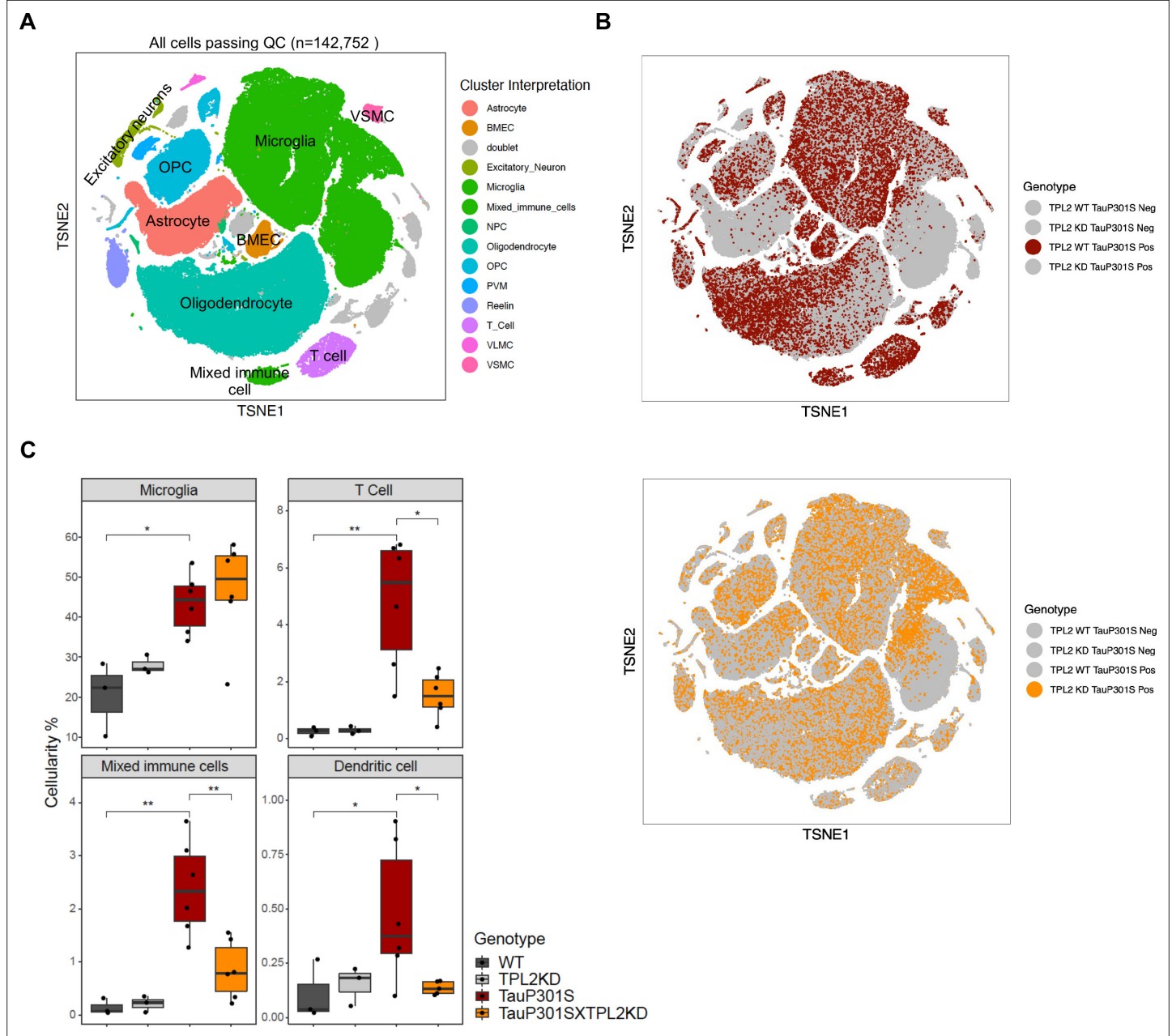

**Figure 5.** Single-cell RNA sequencing (scRNA-seq) shows that tumor progression locus 2 kinase dead (TPL2-KD) normalizes increased T-cells and dendritic cells in TauP301S brains. (**A**) tSNE dimensional reduction and cell type interpretation of 142,752 hippocampal cells from 9-month-old TPL2 WT/KD TauP301S positive/negative mice. (**B**) Distribution of cells by TPL2 genotype in TauP301S positive samples. (**C**) 'Cellularity plot' illustrating proportions of microglia, mixed immune cells, T-cells, and dendritic cells as identified from further analysis of the mixed immune cell cluster described in *Figure 5—figure supplement 3* in each genotype. Each point represents a single animal, and the *y*-axis is the percentage of all cells in the scRNA-seq dataset for that animal of the given cell type. Boxplots showing summarized distribution of the percentage; p-values are based on t-test between indicated groups using ggpubr. *, p<0.05, **, p<0.01.

The online version of this article includes the following figure supplement(s) for figure 5:

**Figure supplement 1.** Dimensional reduction and interpretation of cell clusters by gene set scores.

**Figure supplement 2.** Cellularity plot for all initially annotated cell types and T-cell immunostaining.

**Figure supplement 3.** Identification of a unique cluster within the 'mixed immune cell' cluster that corresponds to dendritic cells.

**Figure supplement 4.** Cellularity plot for all immune cell subclusters.

**Figure supplement 5.** Subclustering of T-cells.

**Figure supplement 6.** Subclustering of endothelial cells identifies a tumor progression locus 2 (TPL2)-dependent subcluster induced by TauP301S.

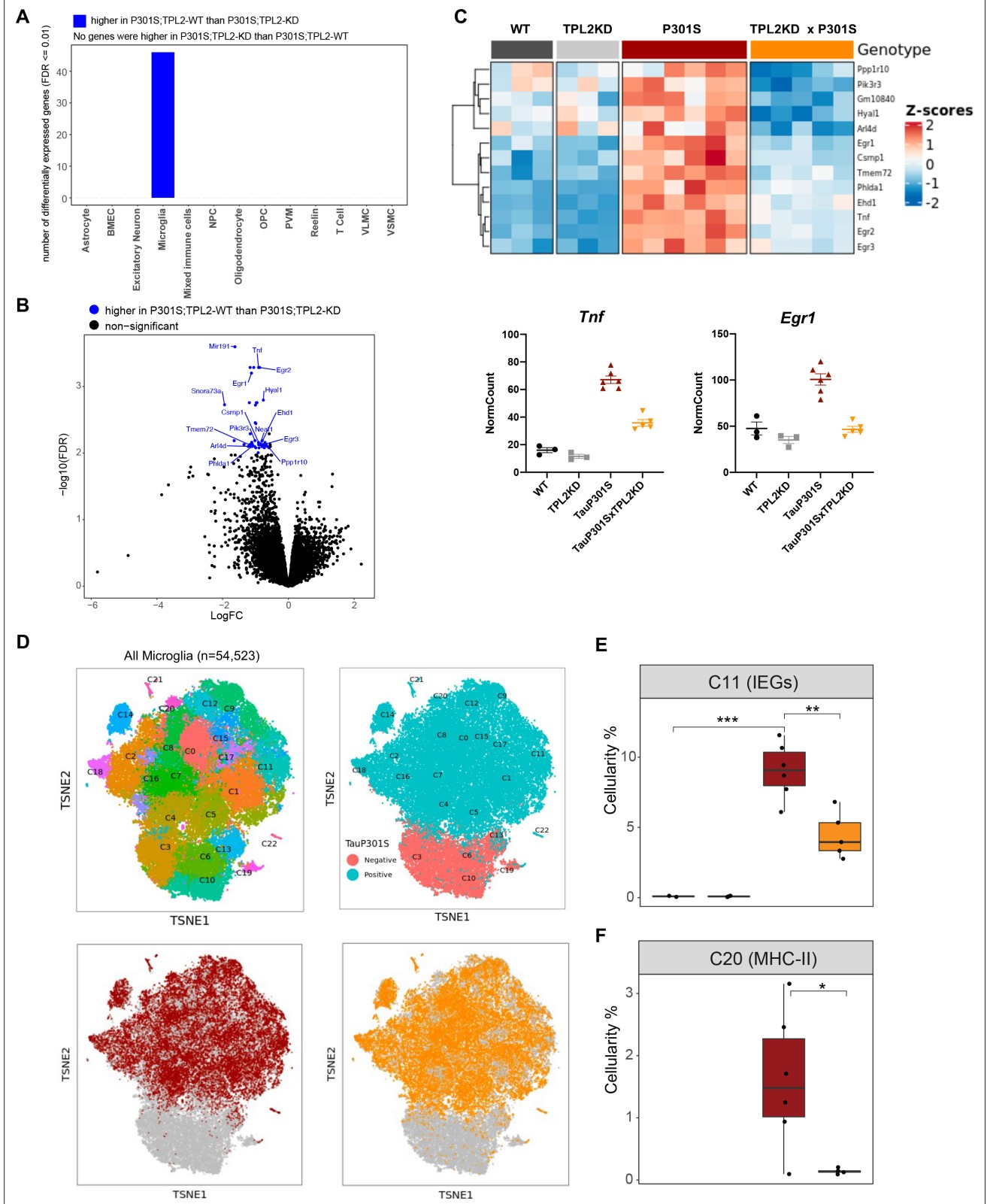

**Figure 6.** Tumor progression locus 2 kinase dead (TPL2-KD) partially normalizes elevated gene expression in TauP301S microglia and reduces the abundance of microglia in specific P301S-induced clusters. (**A**) Number of significantly differentially expressed genes (DEGs) for each cell type from pseudo-bulk analysis at FDR < 0.01 and log2(fold-change) >1.5 or <−1.5 between P301S and P301S;TPL2-KD genotypes. (**B**) Volcano plot showing DEGs for microglia from pseudo-bulk analysis between P301S and P301S;TPL2-KD genotypes. Only protein-coding genes were annotated. (**C**) Heatmap

*Figure 6 continued on next page*

Figure 6 continued

showing z-scores of log2(normCount + 1) for microglia DEGs as shown in (**B**). Boxplots summarizing expression levels of selected genes from the heatmap using normalized counts across the four genotypes used in this study. (**D**) tSNE-based subclustering of n=54,523 microglia defined from *Figure 5A*, colored by subcluster identity and genotypes. (**E and F**) Cellularity plot, similar to *Figure 5C*, showing for each sample the percentage of microglia in subclusters C11 or C20 as a percentage of the total number of microglia cells in that sample. p-Values are based on t-test between indicated groups. *, p<0.05, **, p<0.01.

The online version of this article includes the following figure supplement(s) for figure 6:

**Figure supplement 1.** Subclustering of microglia and interpretation of cell activation states by gene set scores.

**Figure supplement 2.** Cellularity plot for microglia cells.

## TPL2-KD ameliorates dendritic spine loss and brain atrophy in TauP301S mice

Given that TPL2-KD provided neuroprotection in vitro, decreased gliosis, immune cell infiltration, and altered microglia states in P301S mice, we next asked if TPL2-KD provides neuroprotection in P301S mice. As microglia-mediated synapse elimination has been observed in AD mouse models including P301S mice (*Hansen et al., 2018*; *Stephan et al., 2012*; *Wilton et al., 2019*), we quantified synapse density by measuring spine number in sparsely labeled dendrites using GFP expression driven by the Thy1-GFP-M transgene. Spine analysis by ex vivo imaging found that P301S;TPL2-KD mice had significantly increased spine density compared to P301S mice (*Figure 7A*), consistent with potentially reduced microglia removal of spines in P301S;TPL2-KD mice.

Previous studies have shown that TauP301S mice develop neurodegeneration, including ventricle enlargement and brain volume reduction, that can be measured by volumetric MRI (vMRI) (*Dejanovic et al., 2018*; *Wu et al., 2019*). We next investigated if TPL2-KD could protect against neurodegeneration in tauopathy mice using longitudinal vMRI. In WT mice, ventricle volumes stayed stable and neocortex and whole brain volumes increased between 6 and 9 months, and TPL2 kinase deficiency did not alter the trajectory of the brain volume changes. As expected, P301S mice developed brain atrophy, as indicated by increasing ventricle volumes and decreasing brain volumes between 6 and 9 months. Strikingly, TPL2-KD significantly ameliorated the ventricle enlargement and brain volume decline in TauP301S mice (*Figure 7B, C*), indicating a robust neuroprotective effect of inhibition of TPL2 kinase activity in this mouse model. We also measured plasma NfL (neurofilament light chain) to see if this potential biomarker of neurodegeneration was reduced in parallel with the protection against brain volume loss. However, while plasma NfL was strongly increased in P301S mice, TPL2-KD did not significantly affect NfL levels (*Figure 7D*). This indicates that while plasma NfL increases in P301S mice, neuroprotection can be achieved irrespective of NfL levels.

## TPL2-KD normalizes behavioral hyperactivity and memory deficits in TauP301S mice

We next evaluated if the significant attenuation of neuroinflammation and neurodegeneration by TPL2 kinase ablation in tauopathy mice resulted in functional benefits. We first examined locomotor activity in an open field. As previously seen (*Dejanovic et al., 2018*; *Wu et al., 2019*), compared with WT mice, 9-month-old TauP301S mice displayed behavioral hyperactivity as indicated by significantly increased total beam breaks, ambulatory bouts, and rearing behavior. TPL2 kinase activity deficiency did not impact the locomotor activity of non-transgenic mice, but prevented the hyperactivity in the tauopathy mice (*Figure 7E*). To assess the learning and memory functions of these mice, we performed the trace fear conditioning test, a memory test requiring both hippocampus and cortex function (*Burman et al., 2014*; *Han et al., 2003*; *Sharma et al., 2018*). Mice were trained to associate aversive shock with a neutral auditory tone stimulus. When tested 24 hr later, WT mice exhibited elevated freezing behavior in response to the auditory stimulus as expected. P301S mice had somewhat elevated baseline freezing behavior, and failed to exhibit an increase in freezing behavior in response to the auditory stimulus, indicating potential learning and memory deficits. In contrast, P301S;TPL2-KD mice showed normal baseline freezing behavior and elevated freezing behavior in response to the tone stimulus (*Figure 7F*). Thus, in addition to reducing neuroinflammation, and protecting against synapse and neuron loss, eliminating TPL2 kinase activity rescues behavioral alterations in P301S mice.

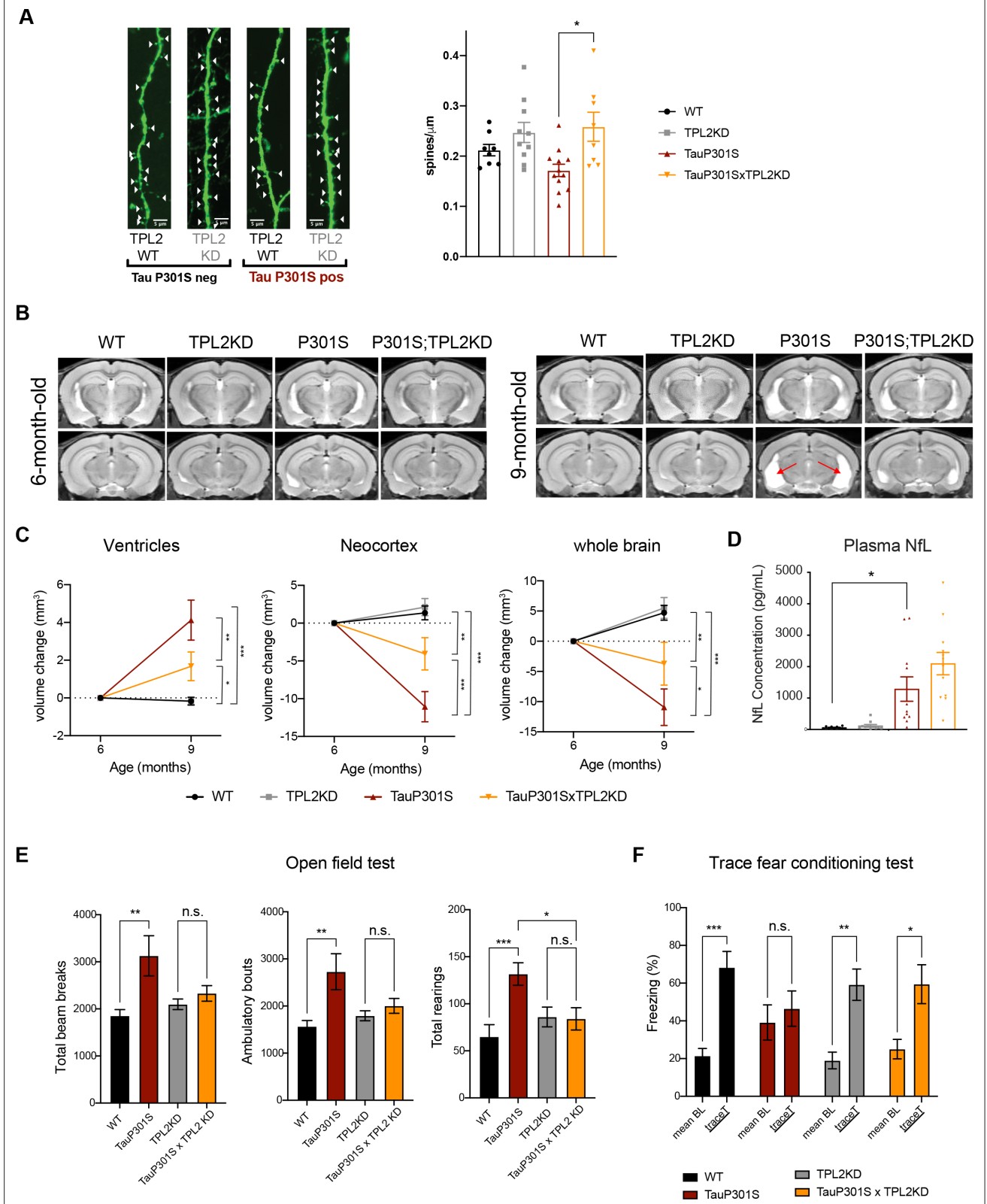

**Figure 7.** Tumor progression locus 2 (TPL2) functional deficiency attenuates brain atrophy and rescues behavioral deficits in TauP301S mice. (A) Ex vivo imaging of fixed mouse hemi-brains of 9-month-old male mice. Representative confocal z-stack images of GFP-expressing neurons are shown on the left with genotypes as indicated. White arrow heads point to the dendritic spines. Scale bar, 5 μm. *Right*, quantification of spine density as shown on the left. Each dot represents one mouse (average of six dendrites). n=8–12. Data are represented by mean ± SEM. *, p<0.05, one-way ANOVA with Tukey's

*Figure 7 continued on next page*

*Figure 7 continued*

multiple comparisons test. (**B**) Representative volumetric MRI images in 6-month-old and 9-month-old male mice with genotypes as indicated. Arrow indicates enlargement of ventricles at 9 months of age compared with 6 months in P301S mice. (**C**) Longitudinal volumetric MRI quantification shows volume changes of ventricles, neocortex, and whole brain at 9 months of age compared with 6 months of age. Comparisons between all four genotypes at 9 months of age showed significant increase of ventricle volumes and decrease of neocortex and whole brain volumes in TauP301S mice and TPL-KD showed partial rescue of the TauP301S-dependent volume changes. n=11–12 male mice per genotype. Data are represented by mean ± SEM. *, p<0.05, **, p<0.01, ***, p<0.001, two-way ANOVA with Tukey's multiple comparisons test. (**D**) Quantification of plasma neurofilament light chain (NfL) levels from mice in (C). (**E**) Spontaneous locomotor activity of mice was evaluated in open field tests including total beam breaks, ambulatory bouts, and total rearings during a 20 min period. TauP301S mice showed hyperactivity, which was rescued by TPL2-KD. n=22–24 animals per genotype with male and female mice combined. *, p<0.05, **, p<0.01, ***, p<0.001, one-way ANOVA with Tukey's multiple comparisons test. (**F**) Memory of mice was evaluated using trace fear conditioning test. Percentage of time freezing during baseline recording time and during the trace interval time after the conditioned stimulus tone was compared for each genotype. n=10–12 animals per genotype. Data are represented by mean ± SEM. *, p<0.05, **, p<0.01, ***, p<0.001, two-way ANOVA with Bonferroni's multiple comparisons test.

## Discussion

Multiple aspects of neuroinflammation have been implicated in both preclinical AD models and AD patient brains, and studies with inhibition of specific neuroinflammation pathways have shown beneficial effects in tauopathy and AD models (*Dejanovic et al., 2018*; *Gate et al., 2020*; *Heneka et al., 2013*; *Hong et al., 2016*; *Ising et al., 2019*; *Laurent et al., 2017*; *Li et al., 2019*; *Liddelow et al., 2017*; *Merlini et al., 2018*; *Ndoja et al., 2020*; *Wu et al., 2019*). Here, we focused on TPL2 which, as a master regulator of the peripheral immune system that is expressed by microglia, could have pleiotropic effects on neuroinflammation. In vitro studies show that similar to its function in the peripheral immune system, TPL2 is also key regulator of microglia in the brain. Pharmacological or genetic inhibition of TPL2 reduced cytokine release and iNOS induction by stimulated microglia in culture and rescued neurodegeneration in a co-culture model. TPL2-KD mice also exhibited strongly ameliorated microglia and astrocyte activation and brain cytokine levels following acute peripheral LPS challenge in vivo.

Based on the potentially beneficial effects of reduced TPL2 kinase function in response to direct stimuli in vitro and in vivo, we tested the effects of TPL2-KD in in vivo neurodegeneration models. That TPL2-KD lessened gliosis but did not provide neuroprotection in ONC and stroke models is perhaps not surprising given the direct severe neuronal damage in these models of acute injury. In contrast, the beneficial effects in the TauP301S mice indicate that modulating immune cell function via eliminating TPL2 kinase activity can be beneficial in a context with chronic neuroinflammation where neurodegeneration gradually occurs over time. This fits with a general model of neuroinflammation in chronic neurodegenerative conditions, where an initial pathology (in this case Tau) results in glial cell activation, and over time activated glial cells contribute to neuronal damage. The partial rescue of brain volume loss in TauP301S mice with TPL2-KD likely corresponds to protection of stressed but functional brain tissue that is being destroyed by the inflammatory response to Tau pathology in the brain. Consistent with this, NfL levels are not rescued with TPL2-KD, which could reflect ongoing NfL production by the protected brain tissue. At the same time it is clear that despite ongoing stress from transgenic Tau production, the increased synapse density and brain volume represent preservation of functional neuronal circuits, as the behavioral phenotypes in P301S were rescued in P301S;TPL2-KD mice, indicating functional benefits.

The tauopathy model study reveals multiple aspects of pathophysiology that are normalized by TPL2-KD concordant with rescue of neurodegeneration and behavior. This includes reductions of the levels of various cytokines, normalization of increased T-cell and dendritic cell abundance in the brain, and reduction in the IEG and MHCII subpopulations of activated microglia. This points to future avenues of research examining which features are of particular importance in the protective effects of TPL2-KD. For example, do specific cytokines play critical roles in pathophysiology? Is an aberrant role for adaptive immunity in the brain critical to degeneration? Are changes in subpopulations of activated microglia critical? Or are the pleiotropic effects of TPL2 kinase deficiency in combination required for benefit? In this context, it is notable that a recent study found T-cell depletion resulted in a robust reduction of neurodegeneration and behavioral deficits in TauP301S mice (*Chen et al., 2023*), suggesting that the reduced amount of T-cells in the brains could be particularly relevant to the protective effects of eliminating TPL2 kinase activity.

While TPL2 is most highly expressed in microglia, it is also expressed at fairly high levels in endothelial cells. Interestingly, a recent study has found that endothelial TPL2 regulates blood-brain barrier permeability and immune cell infiltration in the experimental autoimmune encephalomyelitis (EAE) model (*Nanou et al., 2021*). In that study conditional KO of TPL2 in endothelial cells but not microglia, astrocytes, or neurons, protected against EAE disease severity. Endothelial TPL2 activation was found to allow immune cell infiltration, and this was proposed to function by promoting degradation of tight junction proteins including Cldn5, without changes in their corresponding mRNAs. This raises the possibility that the regulation of T-cell and dendritic cell infiltration into the brain of TauP301S mice could also be mediated by TPL2 function in endothelial cells. Our study utilized a very different model (TauP301S instead of EAE), a different loss-of-function allele (KD, which preserves TPL2 scaffolding function) and we analyzed only RNA and not endothelial proteins. Nevertheless, subclustering analysis of the endothelial cells in our TauP301S scRNA-seq study identified one subcluster (cluster 7) that increased in proportion in TauP301S mice but that was partially normalized in proportion by TPL2-KD in P301S mice (*Figure 5—figure supplement 6A,B*). Cluster 7 specific marker genes included IEGs such as *Ier2/3/5l*, *Fos*, *Ubs*, and *Dusp6* (*Figure 5—figure supplement 6C*) (but not *Cldn5*). While this is consistent with TPL2-dependent regulation of the MAPK signaling pathway in endothelial cells in TauP301S mice, determining whether TPL2 signaling in endothelial cells regulates infiltration of peripheral immune cells in TauP301S mice will require further investigation. Aside from microglia, astrocytes, and endothelial cells in the brain, TPL2 is also expressed in both innate and adaptive immune cells in the peripheral system (*Xu et al., 2018*). Whether and how TPL2 function in various cell types contributes to the disease phenotypes in the tauopathy model or the acute LPS injection model could be investigated in the future by knocking in TPL2-KD in a cell type-specific manner.

Microglia orchestrate both protective and damaging responses during neurodegeneration in conditions such as AD (*Bohlen et al., 2019*; *Hansen et al., 2018*). While our results support a beneficial role of reducing microglia activation by inhibiting TPL2 in a tauopathy model, other recent therapeutic approaches have focused on boosting microglia activation to accelerate clearance of disease-related protein aggregates, such enhancing TREM2-mediated, stimulation of microglia phagocytosis of Aβ (*Fassler et al., 2021*; *Wang et al., 2020*). It would be interesting to investigate the role of TPL2 in a combined transgenic mouse model that has both Aβ and Tau pathology, and evaluate the potential to combine TPL2 inhibition to reduce the harmful aspects of microglia-mediated inflammation with other approaches to boost beneficial effects of microglia activation.

Overall, our data indicate that reducing the inflammatory state of the brain with approaches such as targeting TPL2 kinase activity could be a potential therapeutic approach for diseases with chronic neuroinflammation. Our data also highlight that various TPL2-dependent features of neuroinflammation include elevated brain cytokine level peripheral immune cell infiltration, specific activation states of microglia, etc. that could be explored as other potential therapeutic approaches.

## Materials and methods

**Key resources table**

| Reagent type (species) or resource | Designation | Source or reference | Identifiers | Additional information |
|---|---|---|---|---|
| Strain, strain background (mouse) | TPL2-KD | PMID:17270732 | | |
| Strain, strain background (mouse) | TauP301S | PMID:28420753 | | |
| Antibody | Anti-ERK (1/2) (rabbit polyclonal) | Cell Signalling | Cat. #:9102S; RRID:AB_330744 | 1:1000 |
| Antibody | Anti-phospho-ERK (rabbit monoclonal) | Cell Signaling | Cat. #:4370S; RRID:AB_2315112 | 1:1000 |
| Antibody | Anti-p38 (rabbit polyclonal) | Cell Signaling | Cat. #:9212S; RRID:AB_330713 | 1:1000 |
| Antibody | Anti-phospho-p38 (rabbit polyclonal) | Cell Signaling | Cat. #:9211S; RRID:AB_331641 | 1:1000 |

*Continued on next page*

*Continued*

| Reagent type (species) or resource | Designation | Source or reference | Identifiers | Additional information |
|---|---|---|---|---|
| Antibody | Anti-NF-κB p65 (rabbit monoclonal) | Cell Signaling | Cat. #:8242S; RRID:AB_10859369 | 1:1000 |
| Antibody | Anti-phopho-NF-κB p65 (rabbit monoclonal) | Cell Signaling | Cat. #:3033S; RRID:AB_331284 | 1:1000 |
| Antibody | Anti-iNOS (rabbit monoclonal) | Cell Signaling | Cat. #:13120S; RRID:AB_2687529 | 1:1000 |
| Antibody | Anti-TPL2 (mouse monoclonal) | Santa Cruz Biotechnology | Cat. #:sc-373677; RRID:AB_10915433 | 1:1000 |
| Antibody | Anti-β-Actin (rabbit monoclonal) | Cell Signaling | Cat. #:4970S; RRID:AB_2223172 | 1:5000 |
| Antibody | Anti-calnexin (rabbit polyclonal) | Enzo Lifesciences | Cat. #:ADI-SPA-865-F; RRID:AB_11180747 | 1:1000 |
| Antibody | Anti-CD8a, clone 53–6.7 (rat monoclonal) | BD Biosciences | Cat. #:553027; RRID:AB_394565 | 1:500 |
| Antibody | Anti-CD31 (goat polyclonal) | R&D Systems | Cat. #:AF3628; RRID:AB_2161028 | 1:200 |
| Antibody | Donkey anti-rat Alexa Fluor plus 555 (donkey polyclonal) | Thermo Fisher Scientific | Cat. #: A48270; RRID:AB_2896336 | 1:500 |
| Antibody | Donkey anti-goat Alexa Fluor 647 (donkey polyclonal) | Thermo Fisher Scientific | Cat. #: A-21447; RRID:AB_141844 | 1:500 |
| Antibody | Anti-RBPMS (rabbit polyclonal) | Phosphosolutions | Cat. #: 1830-RBPMS; RRID: AB_2492225 | 1:200 |
| Antibody | Anti-Iba1 (guinea pig polyclonal) | Synaptic Systems | Cat. #: 234004; RRID:AB_2493179 | 1:250 or 1:1000 |
| Antibody | Anti-MAP2, (mouse monoclonal) | Sigma | Cat. #: M9942; RRID:AB_477256 | 1:500 |
| Sequence-based reagent | CD44 TaqMan assay (FAM-MGB) | Thermo Fisher Scientific | Mm01277161_m1 | 1:20 |
| Sequence-based reagent | Gsr TaqMan assay (FAM-MGB) | Thermo Fisher Scientific | Mm00439154_m1 | 1:20 |
| Sequence-based reagent | Hspb6 TaqMan assay (FAM-MGB) | Thermo Fisher Scientific | Mm01176578_m1 | 1:20 |
| Sequence-based reagent | Igfbp3 TaqMan assay (FAM-MGB) | Thermo Fisher Scientific | Mm01187817_m1 | 1:20 |
| Sequence-based reagent | Tspan4 TaqMan assay (FAM-MGB) | Thermo Fisher Scientific | Mm00453538_m1 | 1:20 |
| Sequence-based reagent | GAPDH Taqman assay (VIC-MGB) | Thermo Fisher Scientific | Mm99999915_g1 | 1:20 |
| Peptide, recombinant protein | Interferon gamma | R&D Systems | 485-MI-100/CF | |
| Peptide, recombinant protein | TNFα | R&D Systems | 410-MT-025/CF | |
| Peptide, recombinant protein | IL-1α | R&D Systems | 400-ML-025/CF | |
| Commercial assay or kit | Chromium Single Cell 3' GEM, Library & Gel Bead Kit v3 | 10X Genomics | PN-1000075 | |
| Commercial assay or kit | Chromium Single Cell B Chip Kit | 10X Genomics | PN-1000073 | |
| Commercial assay or kit | Chromium i7 Multiplex Kit | 10X Genomics | PN-120262 | |
| Commercial assay or kit | Neural Tissue Dissociation Kit (P) | Miltenyi Biotech | Cat. #:130-092-628 | |
| Commercial assay or kit | RNeasy Plus Mini Kit | QIAGEN | Cat. #:74134 | |
| Chemical compound, drug | Actinomycin D | Sigma | Cat. #:A1410 | |
| Chemical compound, drug | Lipopolysaccharides | Sigma | Cat. #:L6143 | |
| Chemical compound, drug | Lipopolysaccharides | Sigma | Cat. #:L3024 | |

*Continued on next page*

*Continued*

| Reagent type (species) or resource | Designation | Source or reference | Identifiers | Additional information |
|---|---|---|---|---|
| Software, algorithm | Prism 9 | GraphPad | | |
| Software, algorithm | Fiji | PMID:22743772 | | |
| Software, algorithm | R | R Core Team | | |

## Animal use

Animals were maintained in accordance with the Guide for the Care and Use of Laboratory Animals of the National Institutes of Health. Genentech is an AAALAC-accredited facility and care and handling procedures of animals were reviewed and approved by the Genentech Institutional Animal Care and Use Committee (IACUC) and followed the National Institutes of Health guidelines. Mouse experiments were conducted under protocols 19-0506 and 19-1053 approved by IACUC.

## Animal models

TauP301S mice expressing human Tau with the P301S mutation driven by the PrP promoter (*Yoshiyama et al., 2007*) were crossed to TPL2 kinase-deficient mice carrying D270A mutation generated in-house (*Senger et al., 2017*). All TauP301S mice were heterozygous for the TauP301S transgene and all TPL2-KD mice were homozygous for the TPL2-D270A mutation. For the stroke study, TPL2 kinase-deficient rats carrying D270A mutation generated by Genentech were used. Cohorts were produced with all genotypes as littermates. Experimenters were blind to genotype for all behavioral measurements, microscopic, and histological analyses.

## Human post-mortem brain tissue

Post-mortem brain tissue (superior frontal gyrus) was obtained from patients with putative AD as indicated by cognitive evaluation scores ("EAD") or pathology confirmed AD ("AD") along with age-matched controls from Folio Biosciences. Human samples were procured with Ethics Committee approval and written informed consent. Adult male and female patients aged 63–93 years were used in this study.

## Immunoblotting

Cells were directly lysed in 2× reducing SDS sample buffer. Mouse brain tissues were homogenized in cold RIPA buffer (50 mM Tris-HCl, 150 mM NaCl, 2 mM EDTA, 1% NP-40, 0.1% SDS) supplemented with phosphatase and protease inhibitors, using a TissueLyser (2×30 Hz, 3 min at 4°C; QIAGEN). After homogenization, samples were centrifuged at 20,000 × *g* for 20 min and the supernatants were transferred into new tubes. The supernatants were then boiled in reducing SDS sample buffer. For human brain tissues, after homogenization in RIPA buffer, SDS loading buffer (final 1.5×) without bromophenol blue/DTT was added to the lysates and the lysates were sonicated. Then, the lysates were boiled for 5 min and centrifuged at 15,000 rpm for 15 min. The supernatants were collected. After protein concentrations were measured using BCA reagents (Thermo Fisher), bromophenol blue and DTT were added to the supernatants. The protein samples were separated by Novex Tris-Glycine SDS-PAGE gels (Invitrogen) and transferred to nitrocellulose membranes (Bio-Rad). Membranes were then blocked with 5% milk in TBST or Licor blocking buffer for 1 hr (RT) and incubated with primary antibodies (in 5% milk or 2% milk in TBST or Licor blocking buffer with 0.1% Tween20, 4°C overnight), and then secondary HRP-conjugated antibodies (2 hr RT). Chemiluminescence signals were detected on ChemiDoc (Bio-Rad). Data analysis was done using Image Lab software (Bio-Rad). Antibodies used for western blots are: anti-ERK(1/2) (#9102S; RRID: AB_330744), anti-phospho-ERK (Thr202/Tyr204) (#4370S; RRID: AB_2315112), anti-p38 (#9212S; RRID: AB_330713), anti-phospho-p38 (Thr180/Tyr182) (#9211S; RRID: AB_331641), anti-NF-κB p65 (#8242S; RRID: AB_10859369), anti-phopho-NF-κB p65 (Ser536) (#3033S; RRID: AB_331284), anti-iNOS (clone D6B6S, #13120S; RRID: AB_2687529), anti-β-Actin (#4970S; RRID: AB_2223172). The above antibodies are all from Cell Signaling Technology, Danvers, MA, USA. TPL2 antibody is from Santa Cruz Biotechnology (sc-373677; RRID: AB_10915433). Calnexin antibody is from Enzo Lifesciences (#ADI-SPA-865-F; RRID: AB_11180747).

## In vitro cell cultures

Mouse embryonic cortical neurons were cultured as described previously (*Friedman et al., 2018*). Briefly, cortices from day 15 C57BL/6N embryos (E15) were dissected and washed thrice with pre-cooled Hank's balanced salt solution (HBSS; Invitrogen). Cortical tissue was incubated for 10 min at 37°C in HBSS supplemented with 0.25% trypsin (Invitrogen, CA, USA) and DNase I (Roche CustomBiotech, IN, USA). Tissue was washed thrice with cold HBSS and triturated in plating media containing DNAse I (Gibco Neurobasal Medium [Thermo Fisher Scientific, MA, USA], 20% heat-inactivated horse serum [Thermo Fisher Scientific], 25 mM sucrose, and 0.25% Gibco GlutaMAX [Thermo Fisher Scientific]). Dissociated cells were centrifuged at $125 \times g$ for 5 min at 4°C. Cells were resuspended in the plating medium and plated on poly-D-lysine and laminin-coated glass coverslips in 24-well plates (180K/well). After ~2 hr, the plating medium was replaced with NbAct4 culture medium (Brainbits). Cells were maintained in an incubator at 37°C with 5% $CO_2$ and the medium was renewed using 50% exchange every 3–4 days to maintain cell health.

Primary microglia were cultured from P0-P2 pups as previously described (*Yeh et al., 2016*). Briefly, mouse brains were dissected out and the brain tissue was disrupted by trituration using a 10 mL serological pipette in cold DMEM media. The homogenate was spun at $300 \times g$ for 5 min. The pellet was resuspended in DMEM media and filtered through a 70 µm cell strainer. Dissociated cells were cultured in 175 $cm^2$ flasks with media containing DMEM, 10%FBS, and 1% penicillin/streptomycin. Flasks were rinsed with PBS and new media was added after 24 hr. After ~12 days, microglia were shaken off (125 rpm for 1 hr), collected and added to 1-week-old or 2-week-old mouse cortical neuronal cultures at 1:1 cell number ratio. After 3 hr, neuron and microglia co-cultures were stimulated with 25 ng/mL or 100 ng/mL LPS (Sigma-Aldrich, MO, USA, #L6143) and 30 ng/mL IFNγ. One day after stimulation (1-week-old neuron culture) or 3 days after stimulation (2-week-old neuron culture), cultures were fixed with 4% PFA for 10 min, rinsed with PBS, blocked with 10% goat serum with 0.1% Triton X-100 in PBS, incubated with primary antibodies (in block solution, 4°C, overnight), and then secondary antibodies (in block solution, RT, 1 hr). After staining, cells were mounted using Prolong Diamond Antifade Mountant with DAPI (Thermo Fisher). Primary antibodies used for immunostaining are: anti-MAP2 (Sigma-Aldrich, MO, USA, #M9942; RRID: AB_477256), anti-Iba1 (Synaptic Systems, #234 004; RRID: AB_2493179). Cells were imaged with a confocal laser scanning microscope LSM710 (Carl Zeiss) using Zen 2.3 SP1 software (Carl Zeiss). Maximum intensity projection images were created using ImageJ. Immunostainings were analyzed using MATLAB software.

Primary astrocyte cultures were plated after microglia were shaken off using the primary microglia cell culture protocol. Briefly, microglia were shaken off (125 rpm for 1 hr) and fresh media was added back to the flasks. The flasks were then shaken overnight at 200 rpm and rinsed with PBS twice. Astrocytes were trypsinized, collected, and plated at 75K/well in 48-well plate. Cultures were treated with vehicle, LPS (100 ng/mL), or cytokines (30 ng/mL TNFα [R&D Systems; 410-MT-025/CF]+3 ng/mL IL-1a [R&D Systems; 400 ML-025/CF]) for 24 hr. Culture supernatants were collected and cytokine levels in the samples were measured using a Bio-Plex Pro Mouse Cytokine 23-Plex Assay (Bio-Rad). RNA was extracted using RNAeasy mini kit (QIAGEN) from cell lysates and RT-qPCR was performed using qScript XLT One-Step RT-qPCR ToughMix, Low ROX (QuantaBio).

## Cytokine measurements (Luminex)

For primary microglia culture, after 24 hr incubation with LPS (100 ng/mL) or vehicle control, culture supernatants were collected and cytokine levels in the samples were measured using a Bio-Plex Pro Mouse Cytokine 23-Plex Assay (Bio-Rad).

For in vivo studies, following perfusion with PBS, whole hemi-brains (LPS i.p. injection study) or hippocampi (TPL2xTauP301S study) were isolated and homogenized in cold RIPA buffer as described above for preparation of immunoblotting samples. The protein concentration in each sample (supernatants after centrifugation) was measured by BCA (Thermo Fisher). Cytokine levels in the samples were measured using mouse cytokine 23-Plex Assay (Bio-Rad) or mouse cytokine 32-Plex Assay (Millipore) and normalized to total protein concentration.

## Bulk RNA-seq

Three-month-old WT or TPL2-KD male mice were injected with PBS vehicle control or LPS (10 mg/kg; O111:B4, #L3024, Sigma) (n=5 per condition). 24 hr later, mice were perfused with cold PBS and the

mouse hemi-brains were immediately sub-dissected and preserved in RNA later. RNA was extracted from samples using QIAGEN RNeasy Plus Mini Kit. Concentration of RNA samples was determined using NanoDrop 8000 (Thermo Scientific) and RNA integrity was determined by Fragment Analyzer (Advanced Analytical Technologies). 0.5 µg of total RNA was used as an input material for library preparation using TruSeq RNA Sample Preparation Kit v2 (Illumina). Library size was confirmed using a Fragment Analyzer (Advanced Analytical Technologies). Library concentrations were determined by qPCR-based using a Library quantification kit (KAPA). The libraries were multiplexed and then sequenced on Illumina HiSeq2500 (Illumina) to generate 30 M of single end 50 base pair reads per library (*Srinivasan et al., 2016*).

The fastq sequence files for all RNA-seq samples were filtered for read quality (keeping reads where at least 70% of the cycles had Phred scores ≥ 23), and ribosomal RNA contamination. The remaining reads were aligned to the mouse reference genome (GRCm38) using the GSNAP alignment tool (*Wu and Nacu, 2010*). These steps and the downstream processing of the resulting alignments to obtain read counts were implemented in the Bioconductor package HTSeqGenie (https://bioconductor.org/packages/release/bioc/html/HTSeqGenie.html; *Pau and Reeder, 2023*). Only uniquely mapped reads were used for further analysis. Differential gene expression analysis was performed with voom + limma (*Law et al., 2014*).

For heatmaps (*Figure 3A–C*), gene expression data were first normalized to nRPKM statistic as described (*Law et al., 2014*), then transformed to a log2 scale. Any values less than –40XX were then replaced by 40, and standard z-score calculation was performed (for each gene, subtracting mean and dividing by standard deviation) and then used for visualization.

Gene set scores were calculated as previously described (*Friedman et al., 2018*). Briefly, gene expression values were first transformed (and stabilized) as log2(nRPKM + 1). The gene set score for a sample was then calculated as the average over all genes in the set of log-transformed expression values.

## Single-cell RNA-seq

Nine-month-old WT (n=3), TPL2-KD (n=3), P301S (n=6), or P301S;TPL2-KD (n=6) male mice were perfused with cold PBS and the hippocampi were immediately sub-dissected. Single-cell suspensions were prepared from the hippocampi as described (*Lee et al., 2021a*). Briefly, hippocampi were chopped into small pieces and dissociated with enzyme mixes in a Neural Tissue Dissociation Kit (P) (Miltenyi 130-092-628) in the presence of actinomycin D. After dissociation, cells were resuspended in Hibernate A Low Fluorescence medium (Brainbits) containing 5% FBS, with Calcein Violet AM (Thermo Fisher C34858) and propidium iodide (Thermo Fisher P1304MP). Flow cytometry was used to sort and collect live single-cell suspensions for the scRNA-seq study.

Sample processing and library preparation was carried out using the Chromium Single Cell 3′ Library and Gel Bead Kit v3 (10X Genomics) according to the manufacturer's instructions. Cell-RT mix was prepared to aim 10,000 cells per sample and applied to Chromium Controller for Gel Bead-in-Emulsion (GEM) generation and barcoding. Libraries were sequenced with HiSeq 4000 (Illumina). scRNA-seq data were processed with an in-house analysis pipeline as described (*Lee et al., 2021a*; *Yartseva et al., 2020*). Reads were demultiplexed based on perfect matches to expected cell barcodes. Transcript reads were aligned to the mouse reference genome (GRCm38) using GSNAP (2013-10-10) (*Wu and Nacu, 2010*). Only uniquely mapping reads were considered for downstream analysis. Transcript counts for a given gene were based on the number of unique UMIs (up to one mismatch) for reads overlapping exons in sense orientation. Cell barcodes from empty droplets were filtered by requiring a minimum number of detected transcripts. Sample quality was further assessed based on the distribution of per-cell statistics, such as total number of reads, percentage of reads mapping uniquely to the reference genome, percentage of mapped reads overlapping exons, number of detected transcripts (UMIs), number of detected genes, and percentage of mitochondrial transcripts. After this primary analysis step, cells with less than 1000 total UMIs or greater than 10% mitochondrial UMIs were discarded. UMI normalization was performed by dividing each gene expression value for a cell by a factor proportional to the total number of transcripts in that cell. More precisely put, letting $n_c$ represent the total number of UMIs for cell $c$, then the normalization factor $f_c$ for that cell was given by:

$$f_c = \frac{n_c}{median_{c'}(n_{c'})}$$

(with $c'$ going over all cells) and the 'normalized UMIs' for gene $g$ and cell $c$ given by $nUMI_{g,c} = n_c/f_c$. **Stuart et al., 2019**, was used to calculate PCA, tSNE coordinates, and Louvain clustering for all cells (**Figures 5A, 6D** and **Figure 5—figure supplements 3B, 6A**). Cell-type markers from **Friedman et al., 2018**; **Zeisel et al., 2018**, were used to identify major cell-type clusters, and were interpreted using marker gene sets (**Figure 5A**, **Figure 5—figure supplements 1 and 3B**). Color scales in individual gene tSNE plots indicate gene expression on a log2 scale (log2(normCount + 1)). All gene sets used in this study are available in **Supplementary file 1**.

T-cells were extracted from the dataset, re-normalized and the 3000 most variable genes were identified. Dimensionality reduction on the variable genes was performed using principal component analysis and the top 30 principal components were used as input to Louvain graph-based clustering and UMAP dimensionality reduction. T-cell subtypes were identified based on the marker genes: *Cd3e*, *Cd4*, and *Cd8a*. Additional marker genes were determined using the *scoreMarker* function of the Bioconductor package scran (**Lun et al., 2016**).

A cluster of doublets, co-expressing microglia and T-cell genes, and a cluster of T-cells, only present in two animals and enriched for ribosomal proteins, were identified and excluded from downstream analyses.

T-cell subtype percentages were calculated for each sample either as a percentage of the total number of cells in a sample or as a percentage of the total number of T-cells in a sample.

Pseudo-bulk of various cell type expression profiles were derived from single-cell datasets first by aggregating each sample's data for each cell type as described (**Lee et al., 2021a**; **Lee et al., 2021b**). So, for $n$ samples and $m$ cell types there were $n * m$ total possible pseudo-bulks (i.e. aggregates of cells of a single type from a single sample). If fewer than 10 cells of a particular type were present in a given sample then they were discarded, so the actual total number of pseudo-bulks was less than $n * m$. A single 'raw count' expression profile was created for each pseudo-bulk simply by adding the total number of UMIs for each gene across all cells of that type from that sample. This gave a gene-by-pseudo-bulk count matrix which was then normalized to a normalizedCount statistic using the estimateSizeFactors function from DESeq2 (**Love et al., 2014**), used for calculating gene set scores and visualizing gene expression, and for normalization factors for differential expression (DE) analysis. DE was performed on pseudo-bulk datasets using voom + limma methods for bulk RNA-seq. To put this into more formal notation, let $n_{ij}$ be the raw UMI number of gene $i$ in each cell type $j$. Let $s_j$ indicate the sample of cell $j$. The pseudo-bulk count matrix $B$ with rows indexed by genes and columns indexed by samples (instead of cells) is defined as

$$B_{is} = \sum_{j:s_j \in S} n_{ij}$$

The matrix $B$ is then size-factor normalized and analyzed using the standard methods of bulk RNA-seq, including DE analysis using voom + limma (**Law et al., 2014**). DE analysis was performed mostly as previously described (**Lee et al., 2021a**; **Lee et al., 2021b**; **Srinivasan et al., 2020**) using voom + limma methods for bulk RNA-seq. For each analysis (i.e. for each pair of groups of pseudo-bulks to be compared), low expressed genes were filtered out. This was defined as genes with at least 10 total UMIs in at least three of the analyzed pseudo-bulks. Gene set scores for each pseudo-bulk profile were calculated as described above for scRNA-seq data, except NormCount values were used in the place of nUMI values.

Cellularity (cellular composition) plots are plotted using geom_boxplot (https://ggplot2.tidyverse.org/reference/geom_boxplot.html). The lower and upper hinges correspond to the first and third quartiles (the 25th and 75th percentiles). The upper whisker extends from the hinge to the largest value no further than 1.5 * IQR from the hinge (where IQR is the inter-quartile range, or distance between the first and third quartiles). The lower whisker extends from the hinge to the smallest value at most 1.5 * IQR of the hinge. Data beyond the end of the whiskers are called 'outlying' points and are plotted individually (**Cleveland and McGill, 1985**). p-Values are based on t-test between indicated groups using ggpubr.

## Ex vivo imaging analysis of dendritic spine density

Nine-month-old WT, TPL2-KD, TauP301S, or TauP301S;TPL2-KD male mice that also express Thy1-GFP-M transgene were euthanized, transcardially perfused with PBS, and brains were harvested. The

brains were incubated for 48 hr in 4% PFA, and then washed with PBS and embedded in agarose for ex vivo imaging. Spines on apical dendrites of neurons in somatosensory cortex were imaged with a two-photon laser scanning microscope (Prairie Technologies) using a Ti:sapphire laser (MaiTai DeepSee Spectra Physics; Newport) at 840 nm with a 60× numerical aperture 1.0 objective lens (Olympus). The image resolution is 0.1 mm/pixel across a 1024*1024-pixel FOV using 0.5 mm Z-steps. In each mouse, 6 cells (1 dendrite/cell) were collected. A custom, semiautomated image analysis routines in MATLAB (MathWorks) was used to generate dendritic spine density. The density is calculated as the total number of spines divided by the length of the corresponding dendrite. Imaging and analysis were performed under blinded conditions.

## Longitudinal volumetric brain MRI imaging

MRI was performed on a 7T Bruker (Billerica, MA, USA) system with a four-channel receive-only cryogen-cooled surface coil and a volume transmit coil (Bruker, Billerica, MA, USA). T2-weighted images were acquired with a multi-spin echo sequence: TR 5100 ms, TE 10, 20, 30, 40, 50, 60, 70, 80 ms, 56 contiguous axial slices of 0.3 mm thickness, FOV 19.2 × 19.2 mm$^2$, matrix size 256×128, 1 average, with a scan time of 11 min/mouse. During imaging, anesthesia for mouse was maintained at 1.5% isoflurane and rectal temperature was maintained at 37 ± 1°C using a feedback system with warm air (SA Instruments, Stony Brook, NY, USA). Equal number of males and females were included to detect any gender difference. The regional and voxel differences in the brain structure were evaluated by registration-based region of interest analysis. In brief, multiple echo images were averaged and corrected for field inhomogeneity to maximize the contrast to noise ratio and the images were analyzed based on a 20-region pre-defined in vivo mouse atlas (http://brainatlas.mbi.ufl.edu/) that was co-registered to a study template and warped to individual mouse datasets. All the co-registration steps were performed in SPM8 (Wellcome Trust Centre for Neuroimaging, UCL, UK).

## Tissue processing and IHC

Mice were deeply anesthetized and transcardially perfused with PBS. Hemi-brains were drop fixed for 48 hr at 4°C in 4% paraformaldehyde (PFA) as previously described (*Wu et al., 2019*). After being cryoprotected and frozen, up to 40 hemi-brains were embedded per block in a solid matrix and sectioned coronally at 30 μm (MultiBrain processing by NeuroScience Associates, NSA) before being mounted onto slides. TauP301S × TPL2-KD cohort brain sections were stained for AminoCu, AT8, GFAP, Iba1, CD68, and NeuN at NSA using established protocols. Brightfield slides processed by NeuroScience Associates were imaged on the Nanozoomer system (Hamamatsu Corp, San Jose, CA, USA) at ×200 magnification with a resolution of 0.46 μm/pixel. Quantification of chromogenic staining (AminoCu, NeuN, GFAP, Iba1, AT8, CD68) area was performed using grayscale and color thresholds (*Brey et al., 2003*) followed by morphological operations. Positive stain area was normalized to the whole brain section or manually marked up hippocampal area.

T-cell IHC were performed as previously described (*Lee et al., 2021b*). Brain sections were subject to antigen retrieval by incubating with 10% formic acid for 20 min. After washing in PBS, brain sections were permeabilized and blocked by 0.3% Triton X-100 and 5% BSA in PBS at room temperature for 1 hr. After washing with PBS, brain sections were incubated with primary antibodies (rat anti-CD8 [clone 53–6.7, BD Biosciences 553027] and goat anti-CD31 [R&D Systems AF3628]) in 0.3% Triton X-100 and 1% BSA in PBS overnight at 4°C. After PBS washes, sections were incubated with secondary antibodies (donkey anti-rat Alexa Fluor plus 555 [A48270] and donkey anti-goat Alexa Fluor 647 [A-21447]) for 2 hr at room temperature. After PBS washes, brain sections were mounted onto slides with ProLong Gold Antifade Mountant (Thermo Fisher Scientific P36931). Images were acquired by Zeiss LSM980 (Carl Zeiss) with Plan-Apochromat 20×/0.8 M27 lens. CD8-expressing cells were manually counted using Cell Counter, a built-in plug-in in Fiji. Experimenters were blinded to genotype during all image collection and analysis.

## Open field behavior

Spontaneous locomotor activity of 9-month-old males and 9-month-old females was measured with an automated Photobeam Activity System-Open Field (San Diego Instruments) (*Wu et al., 2019*). Mice were placed individually in a clear plastic chamber (41L × 41W × 38H cm$^3$) surrounded by a loco-motor frame and a rearing frame fitted with 16×16 infrared (IR) beams to record horizontal locomotor

activity (3 cm above the floor) and vertical rearing activity (7.5 cm above the floor), respectively. The total number of beam breaks for both horizontal and vertical movements was calculated for a total of 20 min.

## Trace fear conditioning

10-month-old female mice were individually placed in a fear-conditioning chamber (30 × 24 × 24 cm$^3$ measured inside of the chamber) located inside of a sound-attenuating cubicle (Med Associates). Each chamber was equipped with a house light, an IR light, a speaker that was used to deliver white noise during the training and cued phases of the task, and a near-IR camera to record the movement and/or freezing of the mice. The floors of the compartments were equipped with stainless steel metal grids that were connected to a shock generator. Hardware and data acquisition were controlled by Video Freeze software (Med Associates).

During TFC training, the house lights were on and the metal grids served as the floor. TFC training began with 180 s of baseline recording in which the mice were allowed to explore the chamber, followed by 6 CS-US trials, each consisting of a 20 s tone (90 dB white noise) as the conditioned stimulus, followed by an 18 s trace interval, then a 2 s foot shock (0.5 mA) as the unconditioned stimulus, and finally a 30 s delay. After the six CS-US presentations were completed, the mice were returned to their home cage.

The TFC test was administered 24 hr following training. During TFC testing, the context of the chamber was altered by switching off the house lights, covering the grid floor with a plastic inlay, altering the shape of the chamber walls by another insert, and changing the scent of the chamber with a small amount of 1% acetic acid. Mice had 180 s of baseline recording to explore the new altered context before the CS tone used in training was presented for 20 s, followed by a 20 s trace interval and a 30 s delay. No foot shock was administered during the test. The freezing behavior of the mice during the trace intervals of the training and testing phases was recorded using a near-IR camera and the percentage time freezing was calculated using Video Freeze software (Med Associates).

## Plasma NfL measurements

NfL and p-NfH were measured by Simoa platform at Quanterix, Inc, Billerica, MA, USA, using NF-Light Advantage Kit (#103186) and pNF-heavy Discovery Kit (#102669).

## ONC model

ONC was performed as previously described (*Watkins et al., 2013*). About 3-month-old WT or TPL2-KD male mice were used. The left eye of each animal was subjected to ONC injury via an intra-orbital approach under deep anesthesia. An incision was made on the superior conjunctiva of the surgery eye, and the optic nerve was exposed. The crush injury was inflicted for 1 s using a 45° angled fine-tip forceps 1–2 mm from the eyeball. Unoperated right eyes were used as controls.

Whole-mount retina IHC was performed as previously described (*Watkins et al., 2013*). Mouse eyes were collected and fixed in 4% PFA at room temperature for 1–2 hr. After two washes in PBS, retinas were dissected out and permeabilized in PBST (0.5% Triton X-100) for 1 hr. The retinas were then incubated in blocking buffer (5% donkey serum and 0.5% Triton X-100 in PBS) at 4°C overnight, followed by overnight incubation at 4°C in primary antibodies, including anti-RBPMS (Phosphoso-lutions, #1830-RBPMS) and anti-Iba1 (synaptic systems, #234 004) diluted in blocking buffer. After washing in PBST for 6 hr at 4°C (six washes; 1 hr each), the retinas were incubated overnight at 4°C with secondary antibodies and washed with PBST for 6 hr. Before mounting on slides, each retina was rinsed in PBS and was cut with four to five slits along the radial axes from the edge to about two-thirds of the radius to flatten the retina and then mounted on slides. Number of RGCs and microglia were counted using MATLAB software.

## Rat stroke model

Rat stroke model was performed as previously described (*Stark et al., 2021*). All animal experiments were performed as specified in the license authorized by the National Animal Experiment Board of Finland (Eläinkoelautakunta, ELLA) and according to the National Institutes of Health (Bethesda, MD, USA) guidelines for the care and use of laboratory animals. In total, 60 adult male rats (30 WT and 30 TPL2-KD) were generated through intercrossing and assigned to the study at the age of ~9–10 weeks

of age. Transient focal cerebral ischemia was produced by middle cerebral artery (MCA) occlusion in right hemisphere of brain in male rats. The rats were anesthetized with 5% isoflurane. During the operation, the concentration of anesthetic was reduced to 1.0–1.5%. After midline skin incision, the right common carotid artery was exposed, and the external carotid artery was ligated distal from the carotid bifurcation. Filament with silicon-covered tip (Doccol filament 4–0, diameter 0,185 mm, silicon 5–6 mm/diameter 0.35 mm) was inserted 22–23 mm into the internal carotid artery (ICA) up to the origin of MCA. Directly after occlusion, animals were assessed with DWI as described below. After 90 min of ischemia, the MCA blood flow was restored by removal of the thread. The wound was closed and the animals were allowed to recover from anesthesia. Sham animals underwent identical procedures, including anesthesia regime, but without the filament insertion and actual tMCAO.

Lesion size, tissue viability (T2 in ms), and brain edema were determined using absolute T2-MRI on days 2, 15, and 30. MRI acquisition was performed using a horizontal 7T magnet with bore size 160 mm equipped with a gradient set capable of max. gradient strength 750 mT/m and interfaced to a Bruker Avance III console (Bruker Biospin GmbH, Ettlingen, Germany) as described in *Stark et al., 2021*. Multi-slice multi-echo sequence was used with the following parameters; TR = 2.5 s, 12 different echo times (10–120 ms in 10 ms steps) and 4 averages. Eighteen coronal slices of thickness 1 mm were acquired using FOV $30 \times 30$ mm$^2$ and $256 \times 128$ imaging matrix. Blinded volumetric analysis was performed manually, with lesions delineated based on the contrast differences between lesioned and healthy tissues in the ipsilateral side taking into account reference values and contralateral hemisphere as internal control.

A 20-point neuroscore test was used to assess post-tMCAO motor and behavioral deficits at baseline, and on days 1, 3, 7, 14, 21, and 28 post tMCAO. The neurological test was conducted and analyzed in a blinded manner. The following parameters were analyzed: spontaneous circling (max. score 4), motility (max. score 3), general condition (max. score 3), paw placement on table top (max. score 4, 1 point per paw), righting reflex when placed on back (max. score 1), grip strength (max. score 2), contralateral reflex (max. score 1), visual forepaw reaching (max. score 2, 1 for each paw).

At the end-point (D30), the rats were transcardially perfused with heparinized (2.5 IU/mL) saline in order to remove blood from the brains and tissues. After perfusion with heparinized (2.5 IU/mL) saline, mice were perfused with 4% PFA until fixed. Thereafter, whole brains were further immersion fixed in 4% PFA in 0.1 M phosphate buffer (PB) for another 48 hr. Brains were then transferred to 0.1 M PB and stored at 2–4°C until analysis.

IHC was performed at NeuroScience Associates (Knoxville, TN, USA) as described above. The slides were digitally scanned by NeuroScience Associates using the TissueScope LE120 (Huron Digital Pathology, Ontario, Canada) slide scanner at a resolution of 0.40 μm/pixel. Quantitative image analysis was performed with the MATLAB software package (MathWorks). Stained or immunolabeled areas were quantified in each hemisphere and normalized based on the total hemisphere tissue area. At least 27 serial sections were analyzed per stain per mouse and final scores were based on the average of those sections.

## Statistical analysis

Data were analyzed using either GraphPad Prism or R. Statistical testing used for each analysis was listed in the corresponding figure legend. Data are represented by mean ± SEM. Comparisons were considered statistically significant when $p < 0.05$.

## Acknowledgements

We thank members of the Genentech Animal Care staff, fluorescence-activated cell sorting (FACS) laboratory, and CALM and RNA Sequencing laboratory for research support. We thank Dwight Newton for uploading GSE196401 to GEO. We thank Chris Bohlen and Mark Wilson for critical reading of the manuscript.

## Additional information

### Competing interests

Yuanyuan Wang, Tiffany Wu, Ming-Chi Tsai, Mitchell G Rezzonico, Luke Xie, Vineela D Gandham, Hai Ngu, Kimberly Stark, Caspar Glock, Daqi Xu, Oded Foreman, Brad A Friedman, Jesse E Hanson: Employee of Genentech Inc; the author declares no competing financial interests. Alyaa M Abdel-Haleem: Employee of Roche; the author declares no competing financial interests. Morgan Sheng: Former employee of Genentech Inc; the author declares no competing financial interests.

### Funding
No external funding was received for this work.

### Author contributions
Yuanyuan Wang, Conceptualization, Resources, Data curation, Formal analysis, Investigation, Visualization, Methodology, Writing – original draft, Project administration, Writing – review and editing; Tiffany Wu, Resources, Formal analysis, Investigation, Visualization, Methodology, Writing – review and editing; Ming-Chi Tsai, Formal analysis, Investigation, Visualization, Writing – review and editing; Mitchell G Rezzonico, Alyaa M Abdel-Haleem, Vineela D Gandham, Formal analysis, Visualization, Writing – review and editing; Luke Xie, Oded Foreman, Brad A Friedman, Formal analysis, Supervision, Writing – review and editing; Hai Ngu, Formal analysis, Writing – review and editing; Kimberly Stark, Resources, Formal analysis, Project administration, Writing – review and editing; Caspar Glock, Formal analysis, Validation, Writing – review and editing; Daqi Xu, Resources, Writing – review and editing; Morgan Sheng, Conceptualization, Supervision, Writing – review and editing; Jesse E Hanson, Conceptualization, Supervision, Writing – original draft, Project administration, Writing – review and editing

### Author ORCIDs
Yuanyuan Wang ⓘ https://orcid.org/0000-0002-1180-1554
Tiffany Wu ⓘ https://orcid.org/0000-0002-9273-0574
Jesse E Hanson ⓘ https://orcid.org/0000-0003-3059-5132

### Ethics
Human subjects: Human samples were procured with Ethics Committee approval and written informed consent.
Animals were maintained in accordance with the Guide for the Care and Use of Laboratory Animals of the National Institutes of Health. Genentech is an AAALAC-accredited facility and care and handling procedures of animals were reviewed and approved by the Genentech Institutional Animal Care and Use Committee (IACUC) and followed the National Institutes of Health guidelines. Mouse experiments were conducted under protocols 19-0506 and 19-1053 approved by IACUC.

### Decision letter and Author response
Decision letter https://doi.org/10.7554/eLife.83451.sa1
Author response https://doi.org/10.7554/eLife.83451.sa2

## Additional files

### Supplementary files
• Supplementary file 1. Gene sets used to characterize microglia activation states (*Figure 6—figure supplement 1*), cell types (*Figure 5A*), and dendritic cells (*Figure 5—figure supplement 3*).
• MDAR checklist

### Data availability
All data, except for the bulk and single-cell RNAseq, are available in the main text or the supplementary materials. Sequencing data are available on GEO. Accessions numbers are: scRNA-seq including P301S mice: GSE180041; bulk RNA-seq WT, or TPL2-KD mice injected with PBS or LPS: GSE196401; bulk RNA-seq WT, and TauP301S mouse hippocampus: GSE186414.

The following datasets were generated:

| Author(s) | Year | Dataset title | Dataset URL | Database and Identifier |
|---|---|---|---|---|
| Wang Y, Hanson JE, Friedman BA, Rezzonico MG, Mohamed A, Modrusan Z | 2022 | Assessing the role of TPL2 in TauP301S mice | https://www.ncbi.nlm.nih.gov/geo/query/acc.cgi?&acc=GSE180041 | NCBI Gene Expression Omnibus, GSE180041 |
| Wang Y, Hanson JE, Friedman BA, Rezzonico MG, Modrusan Z | 2023 | TPL2 kinase activity regulates microglial inflammatory responses and promotes neurodegeneration in tauopathy mice | https://www.ncbi.nlm.nih.gov/geo/query/acc.cgi?&acc=GSE196401 | NCBI Gene Expression Omnibus, GSE196401 |

The following previously published datasets were used:

| Author(s) | Year | Dataset title | Dataset URL | Database and Identifier |
|---|---|---|---|---|
| Hanson J, Rezzonico M, Friedman B, Wang Y, Mohamed A, Newton D | 2022 | Bulk RNAseq of C1Q knockout TauP301S hippocampi | https://www.ncbi.nlm.nih.gov/geo/query/acc.cgi?&acc=GSE186414 | NCBI Gene Expression Omnibus, GSE186414 |
| Srinivasan K, Friedman BA, Etxeberria A, Huntley MA | 2019 | Alzheimer's gene expression by cell type - SFG | https://www.ncbi.nlm.nih.gov/geo/query/acc.cgi?acc=GSE125050 | NCBI Gene Expression Omnibus, GSE125050 |
| Zhang Y, Chen K, Sloan SA, Bennett ML | 2014 | An RNA-Seq transcriptome and splicing database of neurons, glia, and vascular cells of the cerebral cortex | https://www.ncbi.nlm.nih.gov/geo/query/acc.cgi?acc=GSE52564 | NCBI Gene Expression Omnibus, GSE52564 |
| Srinivasan K, Friedman BA, Larson JL, Lauffer BE | 2016 | LPS and brain inflammation | https://www.ncbi.nlm.nih.gov/geo/query/acc.cgi?acc=GSE75246 | NCBI Gene Expression Omnibus, GSE75246 |
| Friedman BA, Srinivasan K, Ayalon G, Meilandt WJ | 2017 | Tau-P301S sorted cell types | https://www.ncbi.nlm.nih.gov/geo/query/acc.cgi?acc=GSE93180 | NCBI Gene Expression Omnibus, GSE93180 |

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
