## [Editor Report]

In this study, the authors provide important findings supporting a key role for TLP2 as a regulator of neurotoxic and pro-inflammatory cytokine and chemokine release following acute and chronic neuroinflammation. They provide convincing data supporting that the abrogation of TPL2 kinase activity ameliorates disease pathogenesis in a mouse model of tauopathy. This manuscript will be of broad interest to readers in the fields of neuroimmunology and neurodegenerative disease who are interested in the pathogenic effects of innate immune signaling pathways in disease.

---

## [Decision Letter]

**Decision letter after peer review:**

Thank you for submitting your article "TPL2 kinase activity regulates microglial inflammatory responses and promotes neurodegeneration in tauopathy mice" for consideration by *eLife*. Your article has been reviewed by 2 peer reviewers, and the evaluation has been overseen by a Reviewing Editor and Carla Rothlin as the Senior Editor. The following individual involved in review of your submission has agreed to reveal their identity: Christopher K Glass (Reviewer #2).

Essential revisions:

1) Independent confirmation of the population proportions suggested by single cell analysis as suggested by both reviewers

*Reviewer #1 (Recommendations for the authors):*

1. In Figures 2C and 2E, they should specify which inhibitor was used for the iNOS immunoblots and quantification.

2. In the introduction, it will be helpful to also describe the cell expression profile of TPL2 in the brain. In particular, they should highlight that TPL2 is highly expressed by both microglia and endothelial cells in the brains of mice and humans.

3. I do not believe that the Liddelow et al. 2017 manuscript, which they have cited throughout this paper, rigorously demonstrates the harmful effects of activated astrocytes in AD or FTD. They show that activated astrocytes are seen in human neurodegenerative diseases but they do not soundly establish a mechanistic role for activated astrocytes/A1 astrocytes in the pathogenesis of these neurodegenerative disorders. Perhaps there is a better citation to support their claim here.

4. Page 33, line 1075: Currently listed as (D) and (D) when it should be (C) and (D).

5. Figure 7 contains multiple tests and experiments that focused on a single sex. Histology and behavior were also completed at varied timepoints. Can the authors provide complimentary data is the opposing sex or comment as to why both sexes at consistent timepoints were not represented equally across the study?

6. While it is valuable to acknowledge the TREM2 literature and the beneficial effects of increased microglial activation in certain instances, a statement should be included about the complex relationship tauopathies have with the immune system. Harmful and beneficial aspects of microglia-mediated inflammation are very disease-specific as tau and amyloid pathologies can behave in opposing directions to immune activation. To imply TPL2 inhibition will be beneficial in diseases outside of tau pathology is outside the scope of this study. The negative data associated with the stroke and nerve injury experiments support that TPL2 inhibition is not always beneficial.

7. It would be better to provide cytokine and chemokine concentrations, as opposed to relative units, in Figure S3 and Figure 1.

In the legend for Figure 3, it would be helpful if they provided the experimental time point as well as the route and dosage of LPS injection.

*Reviewer #2 (Recommendations for the authors):*

The authors use several mouse models to demonstrate the potential beneficial effects of TLP2 inhibition on microglia reactivity.

– The major point that needs to be addressed relates to the use of snRNA-seq to quantify the number of cell types in the PS31 and crossed TPL2KD mice. Figure 5c suggests that nearly half of the cells recovered from TauP301S mice at 9 months of age are microglia. This seems very unlikely to reflect the actual distribution of microglia even in this severe model of tauopathy. In addition, the number of captured T cells was increased in the PS31 mice and reduced in PS31xTPL2KD mice. Isolating nuclei from brain tissue followed by 10x snRNA-seq may bias capture towards certain cell types. To demonstrate that the number of microglia and T cells are increased in PS31 and reduced in PS31 x TPL2KD the authors need to perform immunohistochemistry studies to confirm the snRNA-seq results (e.g., using for example Iba1 for microglia and CD3, CD4, and CD8 as markers for T cells).

The authors may also wish to address the following points

– Microglia respond rapidly to LPS injection (see Kim et al., Immunity 2021; Savier et al., biorxiv https://doi.org/10.1101/2022.08.04.502805). It is unclear why the authors used a 24h time point to study gene expression. Furthermore, since the authors analyzed bulk RNA-seq, many differentially expressed genes in microglia may be missed due to the low fraction of microglia in brain tissue. The manuscript would be strengthened by purifying microglia and subsequently RNA-seq. The dosing of LPS (10mg/kg) is rather high.

– The authors show in vitro that phosphorylation of ERK and p38 are increased in response to stimulation with the TLR4 agonist LPS and that pharmacological and genetic inhibition of TLP2 kinase reduced phosphorylation of both pERK p38 expression levels. Is phosphorylation of pERK and p38 protein increased in the tau model and after LPS injection in vivo and were their levels reduced in TPL2KD mice?

– In line with previous comments regarding RNA-seq from bulk tissue of LPS-treated mice, the data would have been strengthened if the authors isolated microglia from the hippocampus and performed RNA-seq to determine differentially expressed genes. The number of clusters for microglia (more than 20) is very high in their snRNA-seq from bulk, and only 2 small clusters that are not in detail further characterized are predominantly in PS31 compared to PS31 x TPL2KD mice. It is unclear, how the immediate early gene expression fits in here since different members of the Egr transcription factor family can have pro- or anti-inflammatory effects. The authors do not show differential expression of cytokines like Il1a, Il6, and Ccl2 in their studies with snRNAseq. Could it be that the decreased microglia activation state could be mediated by T cells, which also express TPL2 kinase, rather than microglia?

– The authors show using MRI studies that brain volume loss is reduced in PS31 crossed with TPL2KD. The data on the rescue of cell loss would be strengthened by complementary histological studies. For example, are there changes in the number of neurons (NeuN) in the dentate gyrus of the hippocampus? How do the authors explain that plasma NFL levels are increased in PS31 x TPL2KD mice, as plasma NFL is widely used as a marker for axonal loss/neurodegeneration?

– The authors did not observe changes in the gene expression of astrocytes in PS31 x TPL2KD compared with PS31 mice. This is surprising since it is thought that microglia activate astrocytes (e.g., Liddelow et al., Nature 2017).

– The authors point out that the reduction of iNOS is an important mechanism of reduced neurotoxicity of microglia in vitro in response to LPS. However, iNOS is not measured across in vivo models.

---

## [Author Response]

Essential revisions:1) Independent confirmation of the population proportions suggested by single cell analysis as suggested by both reviewers

We have performed new immunohistochemistry experiments which confirm the changes in T cell population proportions across P301S and TPL2 genotypes suggested by the single cell analysis. This new data is presented in Figure 5—figure supplement 2. Confirmation of total microglial changes between P301S and non-transgenic was already shown by Iba1 immunohistochemistry in Figure 4.

Reviewer #1 (Recommendations for the authors):1. In Figures 2C and 2E, they should specify which inhibitor was used for the iNOS immunoblots and quantification.

Thank you for the suggestion. We added the information in the figure legend.

2. In the introduction, it will be helpful to also describe the cell expression profile of TPL2 in the brain. In particular, they should highlight that TPL2 is highly expressed by both microglia and endothelial cells in the brains of mice and humans.

Thank you for the suggestion. We have highlighted this in the first section of the results where we described expression of TPL2 based on RNAseq data.

3. I do not believe that the Liddelow et al. 2017 manuscript, which they have cited throughout this paper, rigorously demonstrates the harmful effects of activated astrocytes in AD or FTD. They show that activated astrocytes are seen in human neurodegenerative diseases but they do not soundly establish a mechanistic role for activated astrocytes/A1 astrocytes in the pathogenesis of these neurodegenerative disorders. Perhaps there is a better citation to support their claim here.

Thank you for the suggestion. We changed the citation in the manuscript.

4. Page 33, line 1075: Currently listed as (D) and (D) when it should be (C) and (D).

Thank you for the correction.

5. Figure 7 contains multiple tests and experiments that focused on a single sex. Histology and behavior were also completed at varied timepoints. Can the authors provide complimentary data is the opposing sex or comment as to why both sexes at consistent timepoints were not represented equally across the study?

Based on our experience with this TauP301S model, male mice sometimes develop Tau pathology faster than the female mice (Wu et al., 2019), and need to be analyzed separately. Due to time and resources, for some of the measurements, we only performed the experiments in a single sex cohort.

6. While it is valuable to acknowledge the TREM2 literature and the beneficial effects of increased microglial activation in certain instances, a statement should be included about the complex relationship tauopathies have with the immune system. Harmful and beneficial aspects of microglia-mediated inflammation are very disease-specific as tau and amyloid pathologies can behave in opposing directions to immune activation. To imply TPL2 inhibition will be beneficial in diseases outside of tau pathology is outside the scope of this study. The negative data associated with the stroke and nerve injury experiments support that TPL2 inhibition is not always beneficial.

We agree and have edited the discussion.

7. It would be better to provide cytokine and chemokine concentrations, as opposed to relative units, in Figure S3 and Figure 1.

These results are from multiple independent batches of primary microglia cultures. As expected, the overall levels of cytokines secreted from each independent in vitro primary culture vary, and comparisons are done within each experiment. In order to group the data together, the cytokine concentrations are normalized to the LPS stimulation, no TPL2 inhibition, condition for each experiment/culture.

In the legend for Figure 3, it would be helpful if they provided the experimental time point as well as the route and dosage of LPS injection.

We’ve added the information to the figure legend.

Reviewer #2 (Recommendations for the authors):The authors use several mouse models to demonstrate the potential beneficial effects of TLP2 inhibition on microglia reactivity.– The major point that needs to be addressed relates to the use of snRNA-seq to quantify the number of cell types in the PS31 and crossed TPL2KD mice. Figure 5c suggests that nearly half of the cells recovered from TauP301S mice at 9 months of age are microglia. This seems very unlikely to reflect the actual distribution of microglia even in this severe model of tauopathy. In addition, the number of captured T cells was increased in the PS31 mice and reduced in PS31xTPL2KD mice. Isolating nuclei from brain tissue followed by 10x snRNA-seq may bias capture towards certain cell types. To demonstrate that the number of microglia and T cells are increased in PS31 and reduced in PS31 x TPL2KD the authors need to perform immunohistochemistry studies to confirm the snRNA-seq results (e.g., using for example Iba1 for microglia and CD3, CD4, and CD8 as markers for T cells).

Thank you for the comment. We isolated live cells from brain tissue for scRNAseq. While this method does not capture the actual distribution of cell types (for example neurons are underrepresented compared to snRNAseq experiments), relative comparisons between genotypes can be made. Regarding T cells, we have now performed new IHC experiments quantifying T cell abundance. These new results corroborate the single cell data by empirically showing significantly increased numbers of T cells in TauP301S mice and significantly reduced numbers in the TauP301S x TPL2KD mice (Figure 5—figure supplement 2). Regarding IHC confirmation of the microglial changes, this is shown with IBA1 staining in Figure 4A,B. Consistent with single-cell RNAseq data, IBA1 staining was significantly increased in TauP301S mice but no statistically significant effects of TPL2 genotype were observed.

The authors may also wish to address the following points– Microglia respond rapidly to LPS injection (see Kim et al., Immunity 2021; Savier et al., biorxiv https://doi.org/10.1101/2022.08.04.502805). It is unclear why the authors used a 24h time point to study gene expression. Furthermore, since the authors analyzed bulk RNA-seq, many differentially expressed genes in microglia may be missed due to the low fraction of microglia in brain tissue. The manuscript would be strengthened by purifying microglia and subsequently RNA-seq. The dosing of LPS (10mg/kg) is rather high.

We agree that different microglial genes may have different time courses of response to LPS injection. We chose 24hr time point based on previous publication showing robust microglial gene expression changes (Srinivasan et al., 2016) and astrocyte gene expression changes (Liddelow *et al.*, 2017; Zamanian et al., 2012) at this time point. One advantage of the bulk RNAseq is that although the sensitivity in a particular cell type is reduced, we could capture changes in multiple cell types more conveniently. As shown in Figure 3, we captured many LPS- and TPL2-dependent DEGs which are expressed by microglia or astrocytes, which provided useful insight. We selected this dose based on Srinivasan et al., 2016, although we note that LPS from different vendor/lots may have different potency, so it is hard to compare doses across studies.

– The authors show in vitro that phosphorylation of ERK and p38 are increased in response to stimulation with the TLR4 agonist LPS and that pharmacological and genetic inhibition of TLP2 kinase reduced phosphorylation of both pERK p38 expression levels. Is phosphorylation of pERK and p38 protein increased in the tau model and after LPS injection in vivo and were their levels reduced in TPL2KD mice?

We attempted western blots from mouse brain lysates of non-transgenic and TauP301S mice and observed no increase of p-ERK and p-p38 signal in TauP301S mice. Since ERK and p38 are highly expressed in neurons in the mouse brain, we suspect that the signals were most likely dominated by p-ERK and p-p38 from neurons, masking any potential changes in microglia.

– In line with previous comments regarding RNA-seq from bulk tissue of LPS-treated mice, the data would have been strengthened if the authors isolated microglia from the hippocampus and performed RNA-seq to determine differentially expressed genes. The number of clusters for microglia (more than 20) is very high in their snRNA-seq from bulk, and only 2 small clusters that are not in detail further characterized are predominantly in PS31 compared to PS31 x TPL2KD mice. It is unclear, how the immediate early gene expression fits in here since different members of the Egr transcription factor family can have pro- or anti-inflammatory effects. The authors do not show differential expression of cytokines like Il1a, Il6, and Ccl2 in their studies with snRNAseq. Could it be that the decreased microglia activation state could be mediated by T cells, which also express TPL2 kinase, rather than microglia?

TPL2-dependent IEG expression changes are consistent with TPL2 being a key regulator of MAPK signaling as IEG can be regulated by MAPK signaling pathways. We agree that how each IEG regulates brain inflammation state remains to be elucidated, and this is probably context dependent. Based on IHC and cytokine measurements from brain lysates (Figure 4), the sum effect of reduction in the IEG and MHCII subpopulations of activated microglia was reduced inflammation in P301SxTPL2KD mouse brain. Whether changes in microglia activation state is cell autonomous, affected by T cells or both, could be addressed in the future by cell type specific KI of TPL2KD. We have added discussion of the possible roles of TPL2 in T cells and other cell types.

– The authors show using MRI studies that brain volume loss is reduced in PS31 crossed with TPL2KD. The data on the rescue of cell loss would be strengthened by complementary histological studies. For example, are there changes in the number of neurons (NeuN) in the dentate gyrus of the hippocampus? How do the authors explain that plasma NFL levels are increased in PS31 x TPL2KD mice, as plasma NFL is widely used as a marker for axonal loss/neurodegeneration?

Based on our experience with this model we consider vMRI as a sensitive measure of neurodegeneration and this readout has the advantage of longitudinal measurements allowing normalization to pre-degeneration time points within animals. In addition to vMRI, we also analyzed the dendritic spines using ex vivo imaging, showing increase of dendritic spines by TPL2KD in Tauopathy mice (Figure 7), implying preservation of functional neuronal circuits.

The lack of reduction of plasma NfL in P301SxTPL2KD mice could reflect ongoing NfL production by the protected neurons that are still stressed by transgenic tau expression but are nonetheless functional (as evidenced by behavioral normalization).

– The authors did not observe changes in the gene expression of astrocytes in PS31 x TPL2KD compared with PS31 mice. This is surprising since it is thought that microglia activate astrocytes (e.g., Liddelow et al., Nature 2017).

We did observe attenuated gene expression changes in astrocytes in TPL2KD mice with the acute LPS injection model, which is more similar to the model used by Liddelow’s group. In contrast, the TauP301S model is a model of chronic neurodegeneration and we only evaluated gene expression profiles at the terminal time point where there is ongoing neurodegeneration and immune cell infiltration, so one possibility is that astrocytes may be chronically activated by stimuli other than microglia cytokines. While we did observe gene expression changes in microglia comparing P301SxTPL2KD with P301S, the TPL2-dependent changes were restricted to subpopulations of microglia so it’s also likely the other populations of activated microglia are sufficient to activate astrocytes.

– The authors point out that the reduction of iNOS is an important mechanism of reduced neurotoxicity of microglia in vitro in response to LPS. However, iNOS is not measured across in vivo models.

We hypothesized reduction of iNOS could be one of the mechanisms that TPL2 inhibition protects against microglia-dependent neuronal damage based on the in vitro data. The mechanisms of reduced neurotoxicity could be model dependent in vivo. We did try to run western blots with bulk brain tissue lysates from TPL2KD x P301S mice but we could not detect any signal. It is possible that the signal could be detected in sorted microglia from mouse brains, which will require fresh mouse brains from an active mouse colony that we no longer have.